# THE REASONING BOUNDARY PARADOX: HOW REINFORCEMENT LEARNING CONSTRAINS LANGUAGE MODELS

## ABSTRACT

Reinforcement Learning with Verifiable Rewards (RLVR) has emerged as a key method for improving Large Language Models' reasoning capabilities, yet recent evidence suggests it may paradoxically shrink the reasoning boundary rather than expand it. This paper investigates the shrinkage issue of RLVR by analyzing its learning dynamics and reveals two critical phenomena that explain this failure. First, we expose *negative interference* in RLVR, where learning to solve certain training problems actively reduces the likelihood of correct solutions for others, leading to the decline of Pass@$k$ performance, or the probability of generating a correct solution within $k$ attempts. Second, we uncover the *winner-take-all* phenomenon: RLVR disproportionately reinforces *problems with high likelihood, correct solutions* under the base model while suppressing other initially low-likelihood ones. Through extensive theoretical and empirical analysis on multiple mathematical reasoning benchmarks, we show that this effect arises from the inherent on-policy sampling in standard RL objectives, causing the model to converge toward narrow solution strategies. Based on these insights, we propose a simple yet effective data curation algorithm that focuses RLVR learning on low-likelihood problems, achieving notable improvement in Pass@$k$ performance.

## 1 INTRODUCTION

Large Language Models (LLMs) have recently shown remarkable capabilities in complex logical tasks such as mathematical reasoning (Cobbe et al., 2021; Hendrycks et al., 2021) and programming (Jimenez et al., 2024; Yang et al., 2024b). A key factor behind this success is *Reinforcement Learning with Verifiable Rewards* (RLVR, Lambert et al. 2025; DeepSeek-AI et al. 2025). RLVR optimizes the LLMs against a binary signal based on objective correctness, eliminating the need for human annotations (OpenAI et al., 2024; MetaAI, 2024b). It is believed that RLVR encourages the emergence of novel reasoning strategies, such as self-reflection and iterative refinement (DeepSeek-AI et al., 2025; Zeng et al., 2025; Luo et al., 2025), enabling the LLMs to surpass the capabilities of their base models.

However, recent studies suggest that RLVR training does not expand the reasoning boundary beyond what the base model already possesses (Yue et al., 2025; Zhao et al., 2025; Zhu et al., 2025; Liu et al., 2025b). Notably, Liu et al. (2025c); Zhao et al. (2025) reveal that the base model already exhibits complex reasoning behaviors even before RLVR, while Yue et al. (2025), using the pass@$k$ metric (i.e., the probability of generating a correct solution within $k$ attempts), demonstrate that RLVR can even shrink the reasoning boundary, reducing the set of problems the model can solve within $k$ trials. Identifying why this coverage shrinkage occurs is crucial for understanding and effectively leveraging RLVR to solve novel problems.

To investigate this issue, this paper analyzes the learning dynamics of RLVR training. Unlike standard RL, which is effective in learning novel strategies within a *single and well-defined Markov Decision Process (MDP)*, in LLMs reasoning, each problem $x$ induces its own MDP with a distinct and unknown reward function $r(x, \cdot)$ (Setlur et al., 2025; Qu et al., 2025). Consequently, learning to solve a problem $x$, defined by one MDP, can affect the ability of the LM to solve another problem $x'$, defined by another MDP. Indeed, our analysis reveals that RLVR is prone to a *negative interference*

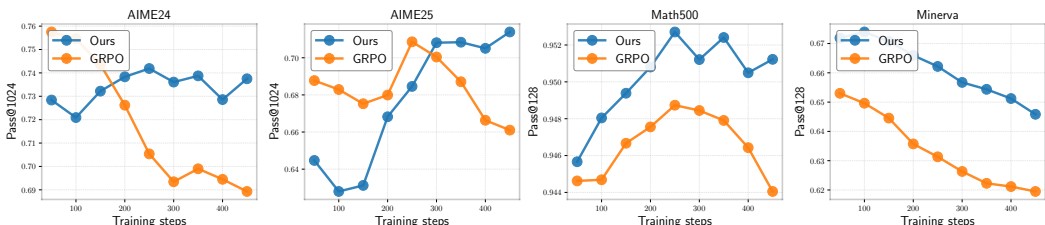

Figure 1: Pass@$k$ evolution (smoothed) during RLVR training with Qwen2.5-Math-1.5B, comparing our proposed finetuning objective SELF to GRPO under a large sampling budget $k$. While GRPO shows a progressive decline, SELF exhibits consistent improvements in Pass@$k$ throughout the training process.

effect where learning to solve a subset of training problems reduces the likelihood of generating correct solutions for others. This dynamic can lead to a *winner-take-all* scenario (Schaul et al., 2019), where the LLMs disproportionately improve on a limited subset of problems that the base model can already solve with high likelihood while neglecting or even regressing on others, ultimately leading to a decline in Pass@$k$ performance or problem coverage. Furthermore, we highlight the failure of current regularization techniques, such as clipping (Schulman et al., 2017) and KL regularization in preventing these effects in RLVR. Finally, these findings motivate us to propose SELF, a simple yet effective data curation algorithm that focuses model learning on a subset of problems with low likelihood, effectively improving the coverage performance across mathematical benchmarks. In summary, the main contributions of this work are as follows:

1. We study the learning dynamics of RLVR and reveal *negative interference* in LLM reasoning, where learning to solve a subset of training problems results in a negative effect on others, leading to *winner-take-all* in which the model only concentrates on solving a smaller set of problems.

2. We reveal the *winner-take-all* learning phenomenon, where RLVR tends to reinforce problems highly solvable under the base model while neglecting problems with initially low likelihood of correct solutions. Due to the existence of *negative interference*, these problems with lower success rates will progressively have lower likelihoods of generating the correct solutions, and on-policy learning cannot provide any explicit learning signal to improve them. This explains the reduction in the diversity of previously learned behaviors during RLVR training or reasoning boundary shrinkage in RLVR.

3. We propose SELF (Selective Examples with Low-likelihood and Forward-KL), a data curation algorithm that selectively solves only problems with low likelihood of arriving at correct answers while preserving previously learned behavior during RLVR. Our extensive empirical evaluation demonstrates that SELF not only improves sample efficiency but also effectively mitigates the coverage shrinkage problem in RLVR.

## 2 RELATED WORKS

**Reinforcement Learning with Verifiable Rewards (RLVR).** RLVR has emerged as a powerful method for improving LLMs in mathematical reasoning and programming (DeepSeek-AI et al., 2025). However, recent evidence suggests that reasoning abilities already exist in base models, with RLVR only amplifying rather than creating such capabilities (Shao et al., 2025; Yue et al., 2025; Liu et al., 2025c). Moreover, RLVR can degrade performance and reduce coverage of correct solutions, evidenced by decreased pass@$k$ metrics (Yue et al., 2025; Dang et al., 2025; Zhao et al., 2025; Wu et al., 2025). Liu et al. (2025a) attributes coverage reduction to limited RL training and overspecialization during pretraining, while Zhu et al. (2025) shows that increasing the likelihood of correct solutions improves accuracy but reduces diversity.

**Loss of Plasticity in Neural Networks.** Neural networks gradually lose adaptability, a phenomenon called *plasticity loss* (Klein et al., 2024; Juliani & Ash, 2024), particularly pronounced in RL due to its non-stationary nature (Klein et al., 2024; Juliani & Ash, 2024; Moalla et al., 2024). Proposed remedies include resetting parameters (Ash & Adams, 2020; Kielo & Lukin, 2024; D'Oro et al.,

2023) and self-distillation (Igl et al., 2021). In particular, Tang & Berseth (2024); Tang et al. (2025) show plasticity loss correlates with drastic model confidence changes (churn), which undermines regularization techniques like clipping (Moalla et al., 2024). Our work explains why RLVR reduces behavioral diversity in LLM reasoning, where non-stationarity arises from both shifting distributions and varying prompt-induced objectives.

**Learning Dynamics.** Ren & Sutherland (2025) shows negative gradients in off-policy LLM fine-tuning create a *squeezing effect*. In RL, *interference* occurs when learning degrades performance on unseen or previously learned states (Liu et al., 2020; 2023). Schaul et al. (2019) demonstrates a *winner-take-all* effect (Schaul et al., 2019; Guo et al., 2018) in contextual bandits, where policies excel in some contexts while regressing in others. Our findings reveal *negative interference* as the key mechanism through which RLVR reduces solvable problems, with on-policy sampling causing disproportionate reinforcement of initially successful problems.

## 3 PROBLEM SETTING AND PER-STEP INFLUENCE IN RLVR

**Language Model.** We first consider a language model (LM) policy $\pi$. For a given prompt $\boldsymbol{x}$, the LM policy $\pi$ will generate a response $\boldsymbol{y}$ in an auto-regressive manner: $\pi(\boldsymbol{y}|\boldsymbol{x}) = \prod_t \pi(y_t|\boldsymbol{x}, \boldsymbol{y}_{<t})$, where $y_t$ is the $t$-th indexed token in $\boldsymbol{y}$ and $\boldsymbol{y}_{<t}$ is the partial completion before $y_t$.

**Reinforcement Learning with Verifiable Rewards (RLVR).** Given a verifiable reward function $r(\boldsymbol{x}, \boldsymbol{y}) \in \{0, 1\}$ indicating whether a response $\boldsymbol{y}$ is correct ($\boldsymbol{y}^+$) or incorrect ($\boldsymbol{y}^-$), the goal of RLVR is to learn a new policy $\pi_\theta$, initialized from a base model (usually the pre-trained model) $\pi_b$, to generate responses maximizing the reward function with the following reward regularized maximization objective:

$$\max_{\pi_\theta} \mathbb{E}_{\boldsymbol{x} \sim \mathcal{D}, \boldsymbol{y} \sim \pi_\theta(\cdot|\boldsymbol{x})} \left[ r(\boldsymbol{x}, \boldsymbol{y}) - \beta \mathrm{KL} \left( \pi_\theta(\cdot|\boldsymbol{x}) \| \pi_b(\cdot|\boldsymbol{x}) \right) \right], \tag{1}$$

where $\beta$ is the KL regularization parameter. Eq. (1) represents a simplified version of common RL algorithms (PPO (Schulman et al., 2017), GRPO (DeepSeek-AI et al., 2025)). We provide a detailed derivation of the GRPO and PPO objectives in Appendix B.

**Per-step Influence.** Learning dynamics describes how the change in the model parameters $\theta$, induced by problem $x$, influences or changes the log-likelihood $\log \pi_\theta$ of other problem-solution pairs $(\boldsymbol{x}', \boldsymbol{y}')$, which can be either training or test examples. Given an update step in LM parameters $\theta^t \to \theta^{t+1}$, the per-step influence measures the change in model confidence $\log \pi_\theta(\boldsymbol{y}'|\boldsymbol{x}')$ on $(\boldsymbol{x}', \boldsymbol{y}')$:

$$\Delta \log \pi_\theta^t(\boldsymbol{y}'|\boldsymbol{x}') = \log \pi_{\theta^{t+1}}(\boldsymbol{y}'|\boldsymbol{x}') - \log \pi_{\theta^t}(\boldsymbol{y}'|\boldsymbol{x}') \tag{2}$$

Using Taylor expansion, assuming with a sufficiently small learning rate $\eta$, we can decompose the per-step influence function as:

**Proposition 3.1.** *Consider a problem-solutions pair $(\boldsymbol{x}, \boldsymbol{y})$ for updating using policy gradient objective, a sufficiently small learning rate $\eta$, the per-step influence on a problem-solution pair $(\boldsymbol{x}', \boldsymbol{y}')$ is defined as:*

$$\Delta \log \pi_\theta^t(\boldsymbol{y}'|\boldsymbol{x}') = \eta \, \mathcal{K}^t(\boldsymbol{x}, \boldsymbol{x}', \boldsymbol{y}, \boldsymbol{y}') A(\boldsymbol{x}, \boldsymbol{y}) + \mathcal{O}(\eta^2) \tag{3}$$

*where $\mathcal{K}^t(\boldsymbol{x}, \boldsymbol{x}', \boldsymbol{y}, \boldsymbol{y}') = \nabla_\theta \log \pi_{\theta^t}(\boldsymbol{y}|\boldsymbol{x})^\top \nabla_\theta \log \pi_{\theta^t}(\boldsymbol{y}'|\boldsymbol{x}')$ measures the influence between $(\boldsymbol{x}, \boldsymbol{y})$ and $(\boldsymbol{x}', \boldsymbol{y}')$, and the advantage function $A(\boldsymbol{x}, \boldsymbol{y})$ determines the direction of the model's update.*

The dot product between two gradients, $\mathcal{K}^t(\boldsymbol{x}, \boldsymbol{x}', \boldsymbol{y}, \boldsymbol{y}')$, can be interpreted as a model-specific similarity measure between problem–solution pairs, and $A(\boldsymbol{x}, \boldsymbol{y})$ determines the *magnitude* and *direction* of the model update, indicating whether the gradient contributes positively or negatively.

## 4 LEARNING DYNAMICS OF RLVR

Calculating $\mathcal{K}^t(\boldsymbol{x}, \boldsymbol{x}', \boldsymbol{y}, \boldsymbol{y}')$ for LLMs is extremely expensive due to their large parameter counts. However, from Proposition 3.1, we have

$$\frac{|\Delta \log \pi_\theta^t(\boldsymbol{y}'|\boldsymbol{x}')|}{\eta \, C} \leq |\mathcal{K}^t(\boldsymbol{x}, \boldsymbol{x}', \boldsymbol{y}, \boldsymbol{y}')|, \tag{4}$$

where $C$ is a constant bounding the advantage function that holds due to finite binary reward values. This bound indicates that we can use the per-step influence $\Delta \log \pi_\theta^t(\boldsymbol{y}'|\boldsymbol{x}')$ to approximate $\mathcal{K}^t(\boldsymbol{x}, \boldsymbol{x}', \boldsymbol{y}, \boldsymbol{y}')$. Given this insight, we introduce two quantitative metrics designed to better characterize the properties of $\mathcal{K}(\boldsymbol{x}, \boldsymbol{x}', \boldsymbol{y}, \boldsymbol{y}')$ as follows.

**Definition 4.1 (Interference in Language Model).** Given a behavior policy $\mu$, we define the interference of a language model policy, $\pi_\theta^t$, at iteration $t$ as:

$$\Delta^+(\pi_{\theta^t}, \mu) = \mathbb{E}_{\substack{\boldsymbol{x}' \sim \mathcal{D}, \\ \boldsymbol{y}^+ \sim \mu(\cdot|\boldsymbol{x})}} \left[ \Delta \log \pi_\theta^t(\boldsymbol{y}^+|\boldsymbol{x}') \right], \tag{5}$$

and the magnitude of the relative changes in model confidence on examples that lie outside of the training batch as:

$$||\Delta(\pi_{\theta^t}, \mu)|| = \mathbb{E}_{\substack{\boldsymbol{x}' \sim \mathcal{D}, \\ \boldsymbol{y} \sim \mu(\cdot|\boldsymbol{x})}} \left( \Delta \log \pi_\theta^t(\boldsymbol{y}|\boldsymbol{x}') \right)^2 .$$

Intuitively, $\Delta^+(\pi_{\theta^t}, \mu)$ measures the *direction* of the relative influence $\mathcal{K}(\boldsymbol{x}, \boldsymbol{x}', \boldsymbol{y}, \boldsymbol{y}^+)$ on other problem–correct solution pairs under the behavior policy $\mu$. A positive $\Delta \log \pi_\theta^t(\boldsymbol{y}^+|\boldsymbol{x})$, indicates improvement, while a negative $\Delta \log \pi_\theta^t(\boldsymbol{y}^+|\boldsymbol{x})$ corresponds to *negative interference*, i.e., learning to solve a training problem $\boldsymbol{x}$ reduces the likelihood of producing other correct solutions. In our work, we consider $\mu$ to be the base model $\pi_b$ to study how RLVR affects the retention (or overwriting) of previously correct problem–solution pairs.

On the other hand, $||\Delta(\pi_{\theta^t}, \mu)||$ measures the *squared magnitude* of the relative influence $\mathcal{K}(\boldsymbol{x}, \boldsymbol{x}', \boldsymbol{y}, \boldsymbol{y})$ – i.e., how drastic the model confidence on examples generated from $\mu$ changes after an update. If $||\Delta(\pi_{\theta^t}, \pi_b)|| = 0$, we can expect no coverage shrinkage to happen during RLVR training, as updating on training examples does not affect other non-training examples. An increase in the magnitude of influence $||\Delta(\pi_{\theta^t}, \pi_b)||$ indicates that the language model struggles to distinguish between different problems, leading to highly correlated gradients across different examples. We provide a detailed explanation in Appendix C.

### 4.1 EXPERIMENTAL SETUP

We focus on mathematical reasoning tasks to investigate the learning dynamics of RLVR. Specifically, we experiment on 2 series of models: Qwen2.5-Math-1.5B/7B (Yang et al., 2024a) and Llama-3.2-3B-Instruct (MetaAI, 2024b). We consider DeepScaleR (Luo et al., 2025) as the training dataset $\mathcal{D}$, which consists of approximately 40k mathematics problems collected from different sources. The detailed experimental setup is provided in Appendix A. We conduct experiments on 4 mathematical reasoning benchmarks: AIME24 & AIME25 (Art of Problem Solving), Math500 (Hendrycks et al., 2021; Lightman et al., 2024), and Minerva (Lewkowycz et al., 2022), following existing works on LLMs reasoning (Yue et al., 2025; Zhu et al., 2025; Li et al., 2025a). Unless otherwise specified, we use $\mathcal{D}_{\text{test}}$ to denote the combined test set from all four benchmarks.

To analyze the model's learning dynamics during training, we create a probing dataset $\mathcal{D}_{\text{prob}}$ (based on the training prompts $\boldsymbol{x} \in \mathcal{D}$) using the base model. For each prompt $\boldsymbol{x}$, we generate 4 responses $\{\boldsymbol{y}_i\}_{i=1}^4$, to balance between computation and solution diversity. We then rely on $\mathcal{D}_{\text{prob}}$ to calculate the effect of interference $\Delta^+$ and the relative change in model confidence on each update $||\Delta(\pi_\theta, \mu)||$ throughout the training process. We present our findings in the following section.

### 4.2 NEGATIVE INTERFERENCE IN RLVR TRAINING OF LANGUAGE MODELS

**Decreasing Pass@$k$ as training progresses.** As shown in Fig. 2(A), RLVR training steadily improves the accuracy (Pass@1) of $\pi_\theta$, averaged across four benchmarks, as training progresses. However, when evaluated with a larger sampling budget (e.g., Pass@256), RLVR displays a clear decline, as the performance drops below that of the base model $\pi_b$ after about 300 steps and continues to deteriorate towards the end of training. The results indicate that RLVR provides little to no improvement in Pass@$k$ at large $k$'s throughout training, a phenomenon consistent with recent findings in (Yue et al., 2025).

**The increase of influence on other examples in one training batch update.** We also observe an increase in $||\Delta(\pi_{\theta^t}, \mu)||$ during training, as shown in Fig. 2(B). The growing strength of influence at each update step on outside examples suggests stronger similarity among gradients across samples

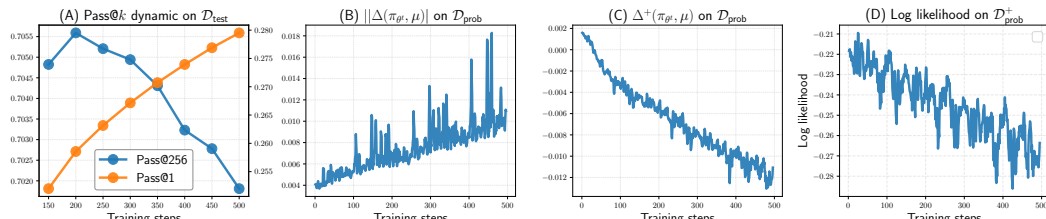

Figure 2: Learning dynamics of RLVR with key trends: (A) RLVR tends to improve average accuracy, but reduce the coverage of solvable solutions, as measured by Pass@256 (averaged across 4 test benchmarks); (B) the increasing effect of influence strength as training progresses; (C) the increase in *negative interference*; and (D) the decline of model confidence on previously correct solutions.

(Tang & Berseth, 2024; Tang et al., 2025), indicating that LLMs struggle to differentiate between distinct problems. As we demonstrate in Sec . 5, this effect can reduce the diversity of previously learned behaviors, where LLMs tend to concentrate on a single reasoning strategy across different problems, and also weaken the regularization effect in RLVR.

**Increasing effect of negative interferences.** We continue to investigate the interference on the other problem-correct solution pairs in Fig. 2(C), and observe a decreasing trend in $\Delta^+(\pi_{\theta^t}, \mu)$; we also observe a similar decreasing trend in the log-likelihood of correct solutions $\log \pi_\theta(\boldsymbol{y}^+|\boldsymbol{x})$ (Fig. 2(D)). These observations indicate that the learned policy increasingly reduces the likelihood of other problem-correct solution pairs generated by the base model. We call this behavior the *negative interference*, where learning to solve certain problems in $\mathcal{D}$ inadvertently hinders the performance on other problem–correct solution pairs.

Later in Sec. 5, we will show that, due to RLVR's on-policy sampling, highly solvable problems in the base model tend to capture most of the learning signal. With *negative inference*, this subset of problems will increasingly dominate the learning process while "weaker" ones are suppressed, ultimately constraining the policy to excel only in this subset of contexts while regressing in others. This also explains why RLVR improves average accuracy but reduces coverage. Under a small sampling budget $k$ (e.g., $k = 1$), RLVR incentivizes these "winner" problems to reach higher average accuracy, but this comes at the cost of *negative interference*, where the model's confidence in correct solutions to other problems is degraded. Consequently, the set of problems solvable under a larger budget $k$ becomes narrower.

**Interference as an indicator of decreasing Pass@$k$.** The previous observations suggest that the decline in Pass@$k$ performance is likely the result of *negative interference* in RLVR learning dynamics. To confirm this hypothesis, we uniformly sample two checkpoints $T$ and $T'$ with $T' > T$ during RLVR training. For each pair, we compute the interference $\Delta^+(\pi_{\theta^{T'}}, \pi_{\theta^T})$ under correct problem-solution pairs of $\mathcal{D}_{\text{test}}$ and the corresponding change in Pass@$k$, i.e., $\Delta \text{Pass@}k = \text{Pass@}k(\pi_{\theta^{T'}}) - \text{Pass@}k(\pi_{\theta^T})$ with $k = 128$ (the value of $k$ is selected based on the related prior works (Yue et al., 2025; Liu et al., 2025a)). We then measure the correlation between these two quantities across four benchmarks.

The correlation results are presented in Fig. 3. As expected, we observe a strong correlation between $\Delta^+(\pi_{\theta^{T'}}, \pi_{\theta^T})$ and the decrease in Pass@$k$, suggesting that interference is a key factor driving the degradation of Pass@$k$ performance during training.

## 5 THE EFFECT OF *on-policy* IN RLVR

In this section, we examine which problems RLVR likely prioritizes. First, we show that RLVR primarily incentivizes the language model $\pi_\theta$ to generate responses that already have high likelihood under the base model, regardless of their correctness. This arises from the nature of *on-policy sampling*: for problems where the base model can produce multiple correct solutions, RLVR tends to reinforce the most probable one; but for those where the correct solutions lie in the "valley" regions of the distribution, RLVR, instead, reinforces the base model's highest-confidence – yet potentially incorrect – responses, offering no meaningful learning signal to escape these incorrect

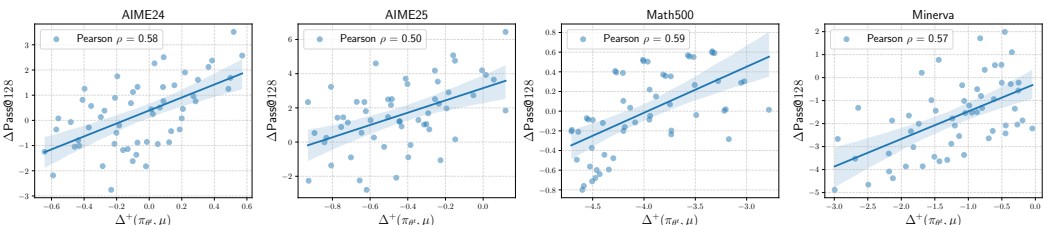

Figure 3: Interference as an indicator of Pass@$k$ decrease across various benchmarks.

modes. Then, we highlight a *winner-take-all* effect that leads to a reduction of previously learned behaviors, as models tend to propagate reasoning strategies from highly solvable problems to others. Finally, we show that existing regularization techniques fail to prevent this issue.

**Experimental Setup.** We consider a subset of problems from the Math500 dataset, denoted as $\mathcal{D}$; the experiments for the other datasets are provided in Appendix H. We utilize the *perplexity* metric (Jurafsky & Martin, 2025):

$$\text{PPL}_\mu(\pi, \mathcal{D}) = \mathbb{E}_{\substack{\boldsymbol{x}\sim\mathcal{D}, \\ \boldsymbol{y}\sim\mu(\cdot|\boldsymbol{x})}} \left[ \exp\left( -\frac{1}{|\boldsymbol{y}|} \log \pi(\boldsymbol{y}|\boldsymbol{x}) \right) \right] \tag{6}$$

where $\mathcal{D} = \{\boldsymbol{x}_i\}_{i=1}^n$ is the set of prompts, the responses are generated by the reference distribution $\boldsymbol{y} \sim \mu(\cdot|\boldsymbol{x})$ conditioned on $\boldsymbol{x}$ ($|\boldsymbol{y}|$ denotes the sequence length), and $\pi$ is the model whose perplexity is being evaluated. We also define the change in average accuracy between the base policy and the learned policy for each problem $\boldsymbol{x}$:

$$\Delta r(\boldsymbol{x}) = \mathbb{E}_{\boldsymbol{y}\sim\pi_\theta(\cdot|\boldsymbol{x})} [r(\boldsymbol{x}, \boldsymbol{y})] - \mathbb{E}_{\boldsymbol{y}\sim\pi_b(\cdot|\boldsymbol{x})} [r(\boldsymbol{x}, \boldsymbol{y})] \tag{7}$$

and then partition $\mathcal{D}$ into two subsets: $\mathcal{D}^\downarrow$, containing problems where RLVR reduces the average accuracy compared to the base model (i.e., $\Delta r(\boldsymbol{x}) < 0$), and $\mathcal{D}^\uparrow$, containing problems where RLVR improves upon the base model (i.e., $\Delta r(\boldsymbol{x}) > 0$). We provide additional experimental details in Appendix H.

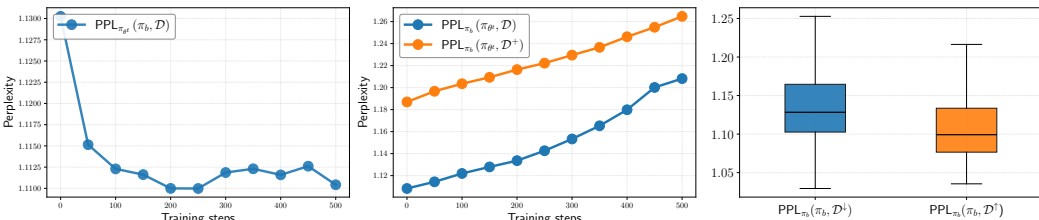

Figure 4: Perplexity during RLVR training. **Leftmost**: the data sampled from each intermediate checkpoint $\pi_{\theta^t}$ exhibits an increasingly high model confidence under $\pi_b$. **Middle**: RLVR models $\pi_{\theta^t}$ exhibit reduced confidence in data previously generated by the base model, regardless of their correctness. **Rightmost**: problems that RLVR improves already have a high likelihood of generating correct solutions under $\pi_b$, while coverage-reduced problems initially have a low likelihood of producing correct answers.

**RLVR tends to increase high likelihood regions under the base model.** Fig. 4 shows that during training, the LM $\pi_\theta$ increasingly generates responses that already have high likelihood under the base model $\pi_b$ (**Leftmost**), while the initial problem-solution pairs sampled from $\pi_b$ increasingly have lower likelihood (or higher $\text{PPL}_{\pi_b}(\pi_{\theta_t}, \mathcal{D})$) in the RLVR policy $\pi_{\theta^t}$ (**Middle**). Together, these results indicate that the sampled responses from $\pi_{\theta^t}$ concentrate on the highest likelihood regions in the base model. We also observe in Fig. 4 (**Middle**) that the problem-correct solution pairs sampled from $\pi_b$ have even lower model confidence (higher $\text{PPL}_{\pi_b}(\pi_{\theta_t}, \mathcal{D}^+)$) and a similar decreasing trend, suggesting that for each problem, RLVR collapses into the solution with the highest likelihood under $\pi_b$ regardless of whether these high likelihood responses is correct or not. Finally, Fig. 4 (**Rightmost**) demonstrates that, for problems where the base model already assigns high likelihood

to correct solutions (lower $\text{PPL}(\pi_{\theta_t}, \mathcal{D}^{\uparrow})$), RLVR reinforces these responses, leading to performance gains (or higher average accuracies). In contrast, when the correct responses initially lie in low-likelihood regions (higher $\text{PPL}(\pi_{\theta_t}, \mathcal{D}^{\downarrow})$), RLVR tends to reduce its probability, reinforcing the base model's initial biases, resulting in degraded performance.

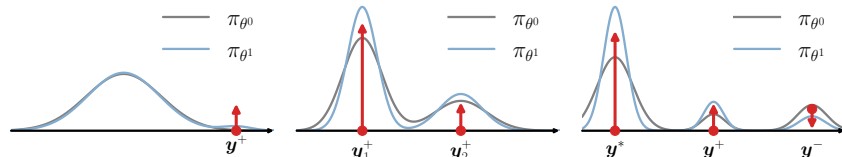

Figure 5: An illustration describing the dynamic of on-policy learning in Eq. (8). **Leftmost**: correct response $\boldsymbol{y}^+$ in low-likelihood regions induce minimal effect. **Middle**: with multiple correct responses $\boldsymbol{y}_1^+$ and $\boldsymbol{y}_2^+$, updates favor the one with higher initial likelihood. **Rightmost**: negative gradients on incorrect response $\boldsymbol{y}^-$ can raise correct ones $\boldsymbol{y}^+$, but greedy responses $\boldsymbol{y}^*$ increase the most.

**Why does RLVR reinforce problems with high-likelihood correct solutions in the base model?** To explain why this phenomenon happens, we first confirm that it occurs when *on-policy sampling* is used for learning (the details are in Appendix D). More specifically, this phenomenon also appears even in a simple bandit problem with a Softmax policy with $V$ actions. We can look into the on-policy objective REINFORCE for an action $\boldsymbol{y}$:

$$\nabla_\theta \mathcal{L}^{\text{REINFORCE}}(\pi_\theta) = A(\boldsymbol{y})\nabla_\theta \pi_\theta(\boldsymbol{y}) = A(\boldsymbol{y})\pi_\theta(\boldsymbol{y})\nabla_\theta \log \pi_\theta(\boldsymbol{y}) \tag{8}$$

and arrive at the following insights (additionally, we utilize Fig. 5 for illustration):

- *Low-likelihood tokens provide less meaningful updates.* Eq. (8) suggests that low likelihood, correct responses receive infrequent and small updates because of the explicit $\pi_\theta(\boldsymbol{y})$ scaling factor and the fact that they are rarely sampled (Fig. 5 **Leftmost**), while high-likelihood tokens dominate the learning process. Moreover, if $\pi_\theta(\boldsymbol{y}) = 0$, the gradient vanishes, creating an additional saddle point where zero-probability tokens cannot be updated.

- *When there are multiple optimal actions $\boldsymbol{y}_1^+$ and $\boldsymbol{y}_2^+$, the one with with the higher likelihood $\pi_\theta(\cdot)$ will dominate.* Due to the scaling of the model probability in Eq. 8, the update will tilt probability toward the already dominant correct mode, as depicted in Fig. 5 (**Middle**). This effect will exacerbate when the gap between the two probabilities increases.

- *Negative gradients tend to reinforce high-likelihood tokens, regardless of their correctness.* On-policy learning with negative gradient exhibits similar behavior to off-policy with negative gradient. When the correct solution resides in a "valley" of the distribution, negative gradient tends to increase high-likelihood tokens regardless of correctness (Fig. 5 **Rightmost**), further diminishing the low-likelihood (but correct) ones (especially when the distribution is peaky). This is also observed in Ren & Sutherland (2025).

- *If $\boldsymbol{y}^+$ is unlikely to be sampled under the $k$-sampling budget $\pi_\theta$, RLVR provides no learning signal.* This arises from the advantage calculation, where $A(\boldsymbol{y}) = 0$ if all sampled actions are incorrect, resulting in no update.

We provide additional theoretical and empirical details of this effect by analyzing the ratio of model predictions $\pi_{\theta^{t+1}}/\pi_{\theta^t}$ in Appendix D.

**On-policy learning and negative interference lead to *Winner-take-all*.** The above analysis suggests that highly solvable problems dominate the learning signal, while problems with lower success rates contribute less due to gradient vanishing. Learning primarily on these highly solvable problems also increases the chance of *negative interference* on problems with lower success rates, and eventually exacerbates the effect of *winner-take-all*; as training progresses, these problems with lower success rates will progressively receive less explicit learning signal and have lower likelihoods of sampling correct solutions.

***Winner-take-all* explains the coverage shrinkage and performance degradation in Minerva Benchmark.** Shao et al. (2025) report that the Qwen2.5-Math model family employs two distinct reasoning modes: code-based and natural-language reasoning. However, during RLVR training, we observe a progressive collapse to natural language reasoning. As shown in Fig. 6 (**Left**), the frequency of the generated solutions using code reasoning steadily decreases over training, while the frequency of natural language reasoning dominates at the end of training.

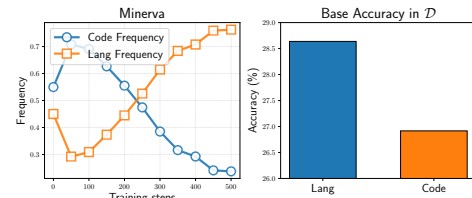

Figure 6: The LM switches from *code reasoning* to *language reasoning* in the Minerva benchmark, due to the reduction of diversity of previously learned behaviors.

We observe that natural language reasoning has a better initial accuracy in the training dataset $\mathcal{D}$ (Fig. 6 (**Right**)). As RLVR training progresses, the LM tends to concentrate on this subset of problems that are already highly solvable under the base model. *Winner-take-all* suggests that RLVR amplifies the highest-likelihood solutions while suppressing other correct ones, leading to reduced diversity among these solvable problems and a collapse to natural language reasoning. As the influence strength $||\Delta(\pi_{\theta^t}, \mu)||$ grows (as shown in Fig. 2), this loss of diversity and the resulting bias towards high-probability responses under base model $\pi_b$ becomes amplified propagate, affecting other problem–solution pairs, including the test problems. Consequently, in cases where the base model's initial bias is incorrect, this self-bias amplification will result in coverage shrinkage and *negative interference*. This explains that on the Minerva benchmark (the test set), while code reasoning achieves higher initial accuracy, as RLVR training gradually switches to language reasoning, the performance on code reasoning problems drops, as observed in Fig. 1.

**Existing regularizations fail to mitigate diversity shrinkage.** In RLVR training, trust region constraints such as clipping aim to prevent the learned policy from deviating too far from the previously updated policy. As shown in Fig. 7, the average probability ratio $\rho = \pi_\theta / \pi_{\text{old}}$ that violates the trust region constraint (lies outside the clipping range $(1 - \epsilon, 1 + \epsilon)$ ($\epsilon = 0.2$)) remains stable in the training batch. However, under the probing dataset $\mathcal{D}_{\text{prob}}$, this violation steadily grows as training progresses. This suggests that the clipping mechanism has little influence on data outside the training batch, thus failing to prevent the *increasing negative interference* throughout the training process. Another regularizer, the Reverse KL, is

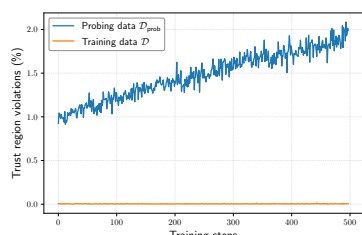

Figure 7: Clipping violations on training data and probing data.

also ineffective, as Reverse KL is known to exhibit mode-seeking behavior, where it has little to no penalty on regions where $\pi_\theta$ puts low probability mass. Furthermore, Reverse KL often favors collapsing probability mass onto high-likelihood mode under the base model $\pi_b$.

## 6 A DATA CURATION TECHNIQUE TO MITIGATE WINNER-TAKE-ALL EFFECT.

Building on the above analysis, we propose a novel and effective algorithm that focuses *learning only on problems where the greedy response fails*. This acts as a proxy metric to exclude highly solvable problems from learning, thus preventing them from dominating the learning signal. To further preserve the diversity of learned behaviors, we replace the Reverse KL regularization with a Forward KL (SFT loss) objective, which penalizes the model when it begins to forget previously learned behaviors. We refer to this method as SELF (**S**elective **E**xamples with **L**ow-likelihood and **F**orward-KL):

$$\mathcal{J}(\pi_\theta) = \mathbb{E}_{\boldsymbol{x} \sim \mathcal{D}, (\boldsymbol{y}^*, \boldsymbol{y}) \sim \pi_\theta(\cdot|\boldsymbol{x})} \left[ \mathbf{1}\{r(\boldsymbol{x}, \boldsymbol{y}^*) \notin \mathcal{C}(\boldsymbol{x})\} r(\boldsymbol{x}, \boldsymbol{y}) - \beta \text{KL}(\pi_b || \pi_\theta) \right] \tag{9}$$

where $\boldsymbol{y}^*$ is the greedy response and $\mathcal{C}(\boldsymbol{x}) = \{\boldsymbol{y} | r(\boldsymbol{x}, \boldsymbol{y}) = 1\}$ is the set of correct completions given a problem $\boldsymbol{x}$.

We compare SELF learning dynamics against the standard GRPO objective in Fig. 8. Since SELF focuses on problems with a low likelihood of being solved correctly, it achieves a lower average training reward than GRPO. However, it preserves better diversity throughout training, as measured by higher token-level entropy across the training process. In addition, SELF effectively mitigates

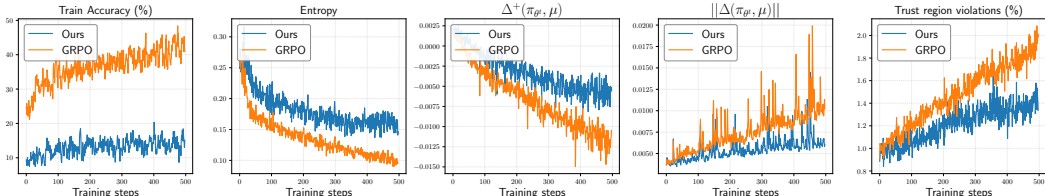

Figure 8: Learning dynamics of GRPO and the proposed method, with key observations: 1) SELF focus on learning problems with low success rate, 2) SELF exhibits better diversity, 3) SELF mitigates the effect of *negative interference* and reduce the influence strength to other examples, result in a strong regularization effect.

the *negative interference effect* as we observe a significant improvement in $\Delta^+(\pi_\theta, \pi_b)$ and a smaller magnitude of influence $||\Delta(\pi_\theta, \mu)||$ on other examples. Furthermore, SELF reduces the extent of trust region violations for examples outside the training batch.

Table 1: Evaluation results on various mathematical benchmarks (%). The best performance under each dataset is marked with **boldface**.

| Model | Method | AIME 2024 | | AIME 2025 | | Math500 | | Minerva | |
|---|---|---|---|---|---|---|---|---|---|
| | | Pass@1 | Pass@1024 | Pass@1 | Pass@1024 | Pass@1 | Pass@256 | Pass@1 | Pass@256 |
| Qwen2.5-Math-1.5B | Base model | 9.73 ±0.10 | 75.86 ±0.4 | 4.45 ±0.07 | 67.80 ±0.56 | 58.83 ±0.11 | 95.58 ±0.51 | 16.82 ±0.03 | **70.67** ±0.51 |
| | GRPO | **15.16** ±**0.06** | 67.77 ±0.55 | **8.50** ±0.3 | 64.05 ±0.35 | **72.67** ±0.05 | 95.0 ±0.07 | 23.96 ±0.01 | 65.07 ±0.18 |
| | SELF | 14.33 ±0.1 | **80.94** ±0.3 | 8.05 ±0.13 | **69.45** ±0.66 | 71.0 ±0.07 | **96.0** ±0.46 | 23.14 ±0.1 | 68.75 ±0.48 |
| Qwen2.5-Math-7B | Base model | 16.94 ±0.41 | 81.83±0.75 | 6.97 ±0.27 | 67.57 ±1.21 | 59.71 ±0.036 | 96.0 ±0.48 | 11.60 ±0.15 | 66.91 ±0.4 |
| | GRPO | **27.95** ±0.61 | 84.77 ±0.72 | 14.13 ±0.36 | 58.60 ±0.96 | **80.36** ±0.18 | 92.8 ±0.14 | 29.40 ±0.13 | 60.66 ±0.14 |
| | SELF | 25.81 ±0.29 | **89.62** ±0.19 | **14.62** ±0.09 | **72.0** ±0.66 | 79.82 ±0.72 | **97.80** ±0.25 | **30.42** ±0.12 | **71.70** ±2.12 |
| Llama-3.2-3B-Instruct | Base model | 3.53 ±0.15 | **53.7** ±0.18 | 0.32 ±0.07 | 48.22 ±0.89 | 26.61 ±0.04 | **91.4** ±1.09 | 8.25 ±0.06 | 49.26 ±0.02 |
| | GRPO | 11.76 ±0.06 | 49.25 ±1.29 | 0.38 ±0.006 | 25.28 ±0.97 | **54.06** ±0.19 | 88.2 ±1.62 | 15.81 ±0.05 | 50.73 ±0.74 |
| | SELF | **13.30** ±0.14 | 51.69 ±0.82 | **0.65** ±0.01 | **51.83** ±0.91 | 52.12 ±0.11 | 91.0 ±0.04 | **17.0** ±0.13 | **57.35** ±0.2 |

Table 1 shows the evaluation of SELF and the baselines on various mathematical benchmarks. SELF achieves comparable Pass@1 performance compared to GRPO, but consistently better performance for larger values of $k$. While GRPO achieves good Pass@1 performance, it also reinforces problems that already have high success rates, leading to narrower coverage at larger $k$. Interestingly, larger models tend to suffer from even more severe coverage shrinkage than smaller models, despite achieving higher Pass@1. In contrast, SELF scales more effectively, outperforming GRPO on Pass@1 in some experiments while also frequently surpassing the base model under larger $k$ budgets. We also provide details of the computational cost of SELF in Appendix G, where SELF demonstrates a negligible computation difference to GRPO.

## 7 CONCLUSION

We investigate why RLVR exhibits coverage shrinkage of solvable problems, indicated by reduced Pass@$k$ performance, compared to the base model, by analyzing its learning dynamics. We identify a detrimental effect, which we call *negative interference*, where learning on a subset of training problems can hinder performance on others. This phenomenon serves as a key indicator of the performance decline in larger $k$ budgets. Furthermore, we find that RLVR tends to concentrate learning on highly solvable problems, leading to reduced diversity and amplification of self-bias toward both previously solvable and unseen problems. Motivated by these findings, we propose SELF, a simple yet effective algorithm that only learn on problems with low success rates, effectively mitigating the coverage shrinkage problem.

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

# A  EXPERIMENTAL DETAILS

## A.1  TRAINING DETAILS

**Models.** We utilize two family of models for RLVR training: Qwen2.5-Math-1.5B and Qwen2.5-Math-7B (Yang et al., 2024a); and Llama-3.2-3B-Instruct (MetaAI, 2024b).

**Training configurations.** We utilize the VERL framework (Sheng et al., 2024) for RLVR training. We train each model on 4 GPUs with a constant learning rate of $10^{-6}$, the maximum response length is 3072 for Qwen2.5-Math models, and 8192 maximum generation tokens for Llama-3.2-3B-Instruct. We use a mini batch size and a rollout batch size of 64. For each prompt, we collect 8 rollouts to compute advantages for the GRPO update. We use a sampling temperature $\tau = 1$. We do not apply Reverse KL divergence or entropy loss in our training, and the total training steps are 500, similar to prior works (Shao et al., 2025; Yu et al., 2025; Luo et al., 2025). We set the hyperparameter controlling the trade-off between reward maximization and Forward KL regularization to $\beta = 10^{-4}$.

**Reward function.** For the reward function $r(\boldsymbol{x}, \boldsymbol{y})$ calculation, we follow the implementation of (Zeng et al., 2025; Hu et al., 2024) with the following rule for faster convergence:

- If the response provides a final answer in the `boxed` format and is correct, it receives a reward of $+1$.
- If the response provides a final answer in the `boxed` format, but it is incorrect, we set the reward as $-0.5$.
- Otherwise, the reward is set to $-1$ for incorrect answers.

## A.2  PROBING DATASET CONSTRUCTION

One major problem when analyzing learning dynamics is the huge response space $\mathcal{Y}$: the number of possible responses $\boldsymbol{y} \in \mathcal{V}^L$, not only includes natural language but also non-sensical texts. In this paper, we are interested in the response region from the base model $\pi_b$, due to the following reason: the concentration coefficient $C(\pi_\theta, \pi_b)$ is small, since RLVR tends to reduce the coverage compared to the base model, we can expect that base model distribution can provide coverage of the learned policy $\pi_\theta$.

To create the probing dataset $\mathcal{D}_{\text{prob}}$, for each problem in the training dataset $\boldsymbol{x} \in \mathcal{D}$, we sample $k = 4$ responses from the base model with temperature $\tau = 0.9$ to ensure diversity and accuracy.

## A.3  PROMPT TEMPLATE

We use the prompt templates for Qwen2.5-Math models and Llama-3.2-3B-Instruct following (Yang et al., 2024a; Luo et al., 2025), as shown in Table 2 and Table 3.

Table 2: Prompt template for Qwen2.5-Math.

```
<|im_start|>system
Please reason step by step, and put your final answer within \boxed{}.<|im_end|>
<|im_start|>user
{input}
Let's think step by step and output the final answer within \boxed{}.<|im_end|>
<|im_start|>assistant
```

Table 3: Prompt template for Llama-3.2-3B-Instruct.

```
<|begin_of_text|><|start_header_id|>system<|end_header_id|>
Cutting Knowledge Date: December 2023
<|start_header_id|>user<|end_header_id|>
{input}
Let's think step by step and output the final answer within \boxed{}.<|eot_id|>
<|eot_id|><|start_header_id|>assistant<|end_header_id|>
```

### A.4 EVALUATION DETAILS

Across all checkpoints, we evaluate using a temperature $\tau = 0.6$ and a top-$p$ value of 0.95, following prior works (Yue et al., 2025; Zeng et al., 2025; Lewkowycz et al., 2022), with a maximum generation of 3072 tokens for Qwen2.5-Math due to limited context length. For Llama-3.2-3B-Instruct, we allow the model to generate a maximum of 16384 tokens.

## B DETAILS DERIVATION OF GRPO

Given a problem $x$ and the generated solutions $\{y\}_{i=1}^{G}$, where $G > 2$ is the number of generated solutions per prompt. We define each token index $t$ in the response $y$ as $y_t$ with $1 \le t \le |y|$, where $|y|$ is the sequence length of the reasoning traces. Specifically, for each problem $x$, GRPO objective DeepSeek-AI et al. (2025) will first calculate group-wise normalized advantage function:

$$A(\boldsymbol{x}, \boldsymbol{y}_i) = \frac{r(\boldsymbol{x}, \boldsymbol{y}_i) - \hat{\mu}}{\hat{\sigma}} \tag{10}$$

where $\hat{\mu} = \frac{1}{G} \sum_{j=1}^{G} r(\boldsymbol{x}, \boldsymbol{y}_j)$, and $\hat{\sigma} = \sqrt{\frac{1}{G-1} \sum_{j=1}^{G} (r(\boldsymbol{x}, \boldsymbol{y}_j) - \hat{\mu})^2}$.

Let $\pi_{\text{old}}$ denote the policy from the previous update step that generates the reasoning traces. The probability ratio $\rho_t = \frac{\pi_\theta(y_t | \boldsymbol{x}, \boldsymbol{y}_{<t})}{\pi_{\text{old}}(y_t | \boldsymbol{x}, \boldsymbol{y}_{<t})}$ at each token index $t$, along with PPO clipping mechanism to serves as a proxy of trust region contraint with clipping threshold $\epsilon$, the GRPO objective is defined as:

$$J(\theta) = \mathbb{E}_{\boldsymbol{x} \sim \mathcal{D}, \, \boldsymbol{y} \sim \pi_{\text{old}}(\cdot | x)} \left[ \sum_{t=1}^{|y|} \min \left( \rho_t(y; \theta) \, A(\boldsymbol{x}, \boldsymbol{y}_i), \, \text{clip}(\rho_t(y; \theta), 1 - \epsilon, 1 + \epsilon) A(\boldsymbol{x}, \boldsymbol{y}_i) \right) \right]$$
$$- \beta \, \mathbb{E}_{\boldsymbol{x} \sim \mathcal{D}} \left[ \text{KL}(\pi_\theta(\cdot | \boldsymbol{x}) \, \| \, \pi_b(\cdot | \boldsymbol{x})) \right]. \tag{11}$$

The KL regularization helps the learned policy not deviate too far from the base model. Recently, Yu et al. (2025) suggests that removing KL regularization can provide better performance. Following this, we set $\beta = 0$ to eliminate KL regularization in our work.

## C PER-STEP INFLUENCE PROOFS AND FURTHER ANALYSIS

### C.1 PROOF OF PROPOSITION 3.1

**Proposition C.1.** *Consider a problem-solutions pair $(\boldsymbol{x}, \boldsymbol{y})$ for updating using policy gradient objective, a sufficiently small learning rate $\eta$, the per-step influence on a problem-solution pair $(\boldsymbol{x}', \boldsymbol{y}')$ is defined as:*

$$\Delta \log \pi_\theta^t(\boldsymbol{y}' | \boldsymbol{x}') = \eta \, \mathcal{K}^t(\boldsymbol{x}, \boldsymbol{x}', \boldsymbol{y}, \boldsymbol{y}') A(\boldsymbol{x}, \boldsymbol{y}) + \mathcal{O}(\eta^2) \tag{12}$$

*where $\mathcal{K}^t(\boldsymbol{x}, \boldsymbol{x}', \boldsymbol{y}, \boldsymbol{y}) = \nabla_\theta \log \pi_{\theta^t}(\boldsymbol{y} | \boldsymbol{x})^\top \nabla_\theta \log \pi_{\theta^t}(\boldsymbol{y}' | \boldsymbol{x}')$ measures the influence between $(\boldsymbol{x}, \boldsymbol{y})$ and $(\boldsymbol{x}', \boldsymbol{y}')$, and the advantage function $A(\boldsymbol{x}, \boldsymbol{y})$ determines the magnitude and direction of the model's update.*

*Proof.* We first approximate $\log \pi_{\theta^{t+1}}(\boldsymbol{y}|\boldsymbol{x})$ using Taylor expansion. For simplicity, we denote $\pi_{\theta^t} = \pi_\theta^t$:

$$\log \pi_\theta^{t+1}(\boldsymbol{y}|\boldsymbol{x}) = \log \pi_\theta^t(\boldsymbol{y}|\boldsymbol{x}) + \nabla_\theta \log \pi_\theta^t(\boldsymbol{y}|\boldsymbol{x})^\top (\theta^{t+1} - \theta^t) + \mathcal{O}(\eta^2) \tag{13}$$

Using the definition of stochastic gradient ascent with a sufficiently small learning rate $\eta$, we have that:

$$\theta^{t+1} = \theta^t + \eta \nabla_\theta \mathcal{J}(\pi_\theta) \tag{14}$$

where $\nabla_\theta \mathcal{J}(\pi_\theta) = A(\boldsymbol{x}, \boldsymbol{y}) \nabla_\theta \log \pi_\theta(\boldsymbol{y}|\boldsymbol{x})$ is the standard policy gradient objective (Qu et al., 2025; Ahmadian et al., 2024), we plug in the above objective in Eq. 13:

$$\Delta \log \pi_\theta^t(\boldsymbol{y}'|\boldsymbol{x}') = \eta \mathcal{K}(\boldsymbol{x}, \boldsymbol{x}', \boldsymbol{y}, \boldsymbol{y}') A(\boldsymbol{x}, \boldsymbol{y}) + \mathcal{O}(\eta^2) \tag{15}$$

which concludes the proof. □

## C.2 Interpretation of the relative strength of influence $||\Delta(\pi_{\theta^t}\mu)||$

If we view the language model $\pi_\theta$ as a neural network with a problem and solution pair $\boldsymbol{o} = (\boldsymbol{x}, \boldsymbol{y})$ as an input, where the output is the log-probability

$$\log \pi_\theta(\boldsymbol{y}|\boldsymbol{x}) = \frac{1}{|\boldsymbol{y}|} \sum_{t=1}^{|\boldsymbol{y}|} \log \pi_\theta(y_t|\boldsymbol{x}, \boldsymbol{y}_{<t}) \tag{16}$$

where $\boldsymbol{y}_{<t}$ is the partial completion before the token index $t$. We consider the gradient of $\pi_\theta$ with respect to parameter $\theta$ at $(\boldsymbol{x}, \boldsymbol{y})$ as $\nabla_\theta \log \pi_\theta(\boldsymbol{y}|\boldsymbol{x}) \in \mathbb{R}^d$ where $d$ is the parameter dimension. We can define a matrix of gradient dot products $\mathcal{K}(\boldsymbol{x}, \boldsymbol{x}', \boldsymbol{y}, \boldsymbol{y}') = \mathcal{K}(\boldsymbol{o}, \boldsymbol{o}')$ across all different prompt-response pair:

$$\mathcal{K} = \begin{bmatrix} \mathcal{K}(\boldsymbol{o}_1, \boldsymbol{o}_1) & \mathcal{K}(\boldsymbol{o}_1, \boldsymbol{o}_2) & \cdots \\ \mathcal{K}(\boldsymbol{o}_2, \boldsymbol{o}_1) & \mathcal{K}(\boldsymbol{o}_2, \boldsymbol{o}_2) & \cdots \\ \vdots & \vdots & \ddots \end{bmatrix} = \nabla_\theta \log \pi_\theta(\mathcal{Y}|\mathcal{X})^\top \nabla_\theta \log \pi_\theta(\mathcal{Y}|\mathcal{X}) \tag{17}$$

where $\nabla_\theta \log \pi_\theta(\mathcal{Y}|\mathcal{X}) = [\nabla_\theta \log \pi_\theta(\boldsymbol{y}_1|\boldsymbol{x}_1), \nabla_\theta \log \pi_\theta(\boldsymbol{y}_2|\boldsymbol{x}_2), \cdots]$ denotes the matrix of the gradient of different prompt-solution pairs $(\boldsymbol{x}, \boldsymbol{y})$.

In an idealized scenario, the kernel matrix $\mathcal{K}$ is diagonal, meaning that all off-diagonal terms are zeros, i.e., $\mathcal{K}(\boldsymbol{o}_i, \boldsymbol{o}_j) = 0$ for $i \neq j$. Under this assumption, improving performance on a given training problem-solution pair would not influence any other unseen problems. Consequently, there would be neither *negative interference*, where training on one example degrades performance on others, nor *coverage shrinkage*, where the set of solvable problems becomes narrower.

In practice, however, this assumption rarely holds for language models. Because parameters are shared across all examples, updates to one example inevitably affect others. This manifests as non-zero off-diagonal entries in $\mathcal{K}$, which capture the degree of cross-influence between examples.

The growing magnitude of the relative influence strength, $||\Delta(\pi_{\theta^t}, \mu)||$, can thus be interpreted as evidence that the number and magnitude of these off-diagonal interactions are increasing. Specifically, this signals that the gradients of different examples are becoming more linearly dependent, since the rank of the kernel $\mathcal{K}$ is equivalent to the rank of the gradient matrix $\nabla_\theta \log \pi_\theta(\mathcal{Y}|\mathcal{X})$. This growing entanglement explains why optimization on a subset of problems may inadvertently interfere with generalization to others.

## D Analyzing learning dynamic of on-policy learning in toy bandit problem.

We first consider a contextual bandit problem with a parametrized softmax policy with $V$ actions. where the input data $\boldsymbol{x}$ is a $d$ dimensional feature vector $\boldsymbol{x} \in \mathbb{R}^d$. The policy is defined as a linear

layer with Softmax output:

$$\pi_\theta(y_i|\boldsymbol{x}) = \frac{\exp(z_i(\boldsymbol{x}))}{\sum_{j=1}^{V} \exp(z_j(\boldsymbol{x}))} \tag{18}$$

We consider the objective of REINFORCE and Off-policy maximum likelihood objectives for 2 actions $y_1$ and $y_2$ that receive a positive reward ($r(\boldsymbol{x}, y) = 1$).

$$\mathcal{J}^{\text{REINFORCE}}(\pi_\theta) = \pi_\theta(\boldsymbol{a}_1|\boldsymbol{x}) + \pi_\theta(\boldsymbol{a}_2|\boldsymbol{x}), \quad \mathcal{J}^{\text{MLE}}(\pi_\theta) = \log \pi_\theta(\boldsymbol{a}_1|\boldsymbol{x}) + \log \pi_\theta(\boldsymbol{a}_2|\boldsymbol{x}) \tag{19}$$

Using gradient descent, we can perform parameter update at each update step $t$ with a learning rate $\eta$:

$$\boldsymbol{\theta}^{t+1} = \boldsymbol{\theta}^t + \eta \nabla_{\boldsymbol{\theta}} \mathcal{J}^{\text{REINFORCE}} = \boldsymbol{\theta}^t - \eta \boldsymbol{x} \left[ (\pi_{\theta^t}(y_1)(\pi_{\theta^t}(\boldsymbol{y}) - \boldsymbol{e}(y_1))^\top + \pi_{\theta^t}(y_2)(\pi_{\theta^t}(\boldsymbol{y}) - \boldsymbol{e}(y_2))^\top \right]$$

$$\boldsymbol{\theta}^{t+1} = \boldsymbol{\theta}^t + \eta \nabla_{\boldsymbol{\theta}} \mathcal{J}^{\text{MLE}} = \boldsymbol{\theta}^t - \eta \boldsymbol{x} \left[ ((\pi_{\theta^t}(\boldsymbol{y}) - \boldsymbol{e}(\boldsymbol{y}_1)^\top + (\pi_{\theta^t}(\boldsymbol{y}) - \boldsymbol{e}(\boldsymbol{y}_2))^\top \right] \tag{20}$$

where $\pi_\theta(\boldsymbol{y}) \in \mathbb{R}^V$ is the vector represents probability distribution over all actions, $\boldsymbol{e}(y)$ is a one-hot vector indicates the target action. A key difference between the REINFORCE and MLE objectives is that, in REINFORCE, the probability of the sampled actions directly scales the update step. As a result, actions with higher likelihood receive larger updates, while low-likelihood actions lead to much smaller changes. To analyze how the model's probabilities in each action change, we define a ratio $\alpha_i = \frac{\pi_i^{t+1}}{\pi^{t_i}}$ and use The following lemma describes its behavior:

**Lemma D.1.** *The ratio of confidence change for each $i$ can be represented as:*

$$\alpha_i^{MLE} = \frac{\pi_i^{t+1}}{\pi_i^t} = \frac{\sum_{j=1}^{V} e^{z_j^t}}{\sum_{j=1}^{V} \beta_j e^{z_j^t}}; \quad \alpha_i^{REINFORCE} = \frac{\sum_{j=1}^{V} e^{z_j^t}}{\sum_{j=1}^{V} \gamma_j e^{z_j^t}} \tag{21}$$

*Note that the values of $\beta_j$ also depends on whether $i \in \{y_1, y_2\}$, hence for Case 1 ($i \in \{y_1, y_2\}$), and Case 2 ($i \notin \{y_1, y_2\}$):*

*Case 1:*

$$\beta_j = \begin{cases} e^{-\eta'\left(2(\pi_j^t - \pi_i^t + 1)\right)} & \text{if } j \notin \{y_1, y_2\} \\ 1 & \text{if } j = y_1 \\ e^{-2\eta'(\pi_j^t - \pi_i^t)} & \text{if } j = y_2 \end{cases}; \quad \gamma_j = \begin{cases} e^{-\eta'\left(\Delta\pi^t(\pi_j^t - \pi_i^t) + \pi_i^t\right)} & \text{if } j \notin \{y_1, y_2\} \\ 1 & \text{if } j = y_1 \\ e^{-\eta'\left((\Delta\pi^t - 1)(\pi_j^t - \pi_i^t)\right)} & \text{if } j = y_2 \end{cases} \tag{22}$$

*Case 2:*

$$\beta_j = \begin{cases} e^{-2\eta'\left(\pi_j^t - \pi_i^t\right)} & \text{if } j \notin \{y_1, y_2\} \\ e^{-\eta'\left(2(\pi_j^t - \pi_i^t) - 1\right)} & \text{if } j \in \{y_1, y_2\} \end{cases}; \quad \gamma_j = \begin{cases} e^{-\eta'\Delta\pi^t\left((\pi_j^t - \pi_i^t)\right)} & \text{if } j \notin \{y_1, y_2\} \\ e^{-\eta'\left(\Delta\pi^t(\pi_j^t - \pi_i^t) - \pi_j^t\right)} & \text{if } j \in \{y_1, y_2\} \end{cases} \tag{23}$$

*where $\Delta\pi^t = \pi_{y_1}^t + \pi_{y_2}^t, \eta' = \eta||\boldsymbol{x}||_2^2$.*

*Proof.* We first need to link the logits vector $z^{t+1}$ and $z^t$. From Eq.20, $z^{t+1}$ can be recursively written as:

$$\begin{aligned} z^{t+1} &= (\boldsymbol{\theta}^{t+1})^\top \boldsymbol{x} \\ &= \left(\boldsymbol{\theta}^t + \eta \boldsymbol{x} \nabla_z \mathcal{J})\right)^\top \boldsymbol{x} \\ &= (\boldsymbol{\theta}^t)^\top \boldsymbol{x} + \eta \left(\boldsymbol{x} \nabla_z \mathcal{J}\right)^\top \boldsymbol{x} \\ &= z^t + \eta' \nabla_z \mathcal{J} \end{aligned} \tag{24}$$

where $\eta' = \eta||\boldsymbol{x}||_2^2$. We can write down for each value in vector $z^{t+1}$ for MLE and REINFORCE objectives as:

$$\text{MLE: } z_i^{t+1} = \begin{cases} z_i^t - 2\eta'\pi_i^t & \text{if } i \notin \{y_1, y_2\} \\ z_i^t - 2\eta'\pi_i^t + \eta' & \text{if } i \in \{y_1, y_2\} \end{cases}$$

$$\text{REINFORCE: } z_i^{t+1} = \begin{cases} z^t - \eta'(\Delta\pi^t \cdot \pi_i^t) & \text{if } i \notin \{y_1, y_2\} \\ z^t - \eta'(\Delta\pi^t - 1)\pi_i^t & \text{if } i \in \{y_1, y_2\} \end{cases}$$

Then, we can write down the probability change $\pi_i^{t+1}$ for each action $i$. For case $i \in \{y_1, y_2\}$, we have the following derivation for MLE objective:

$$\pi_{y_1}^t = \frac{e^{z_i^{t+1}}}{\sum_{j=1}^{V} e^{z_j^{t+1}}} = \frac{e^{z_i^t - 2\eta' \pi_i^t + \eta'}}{\sum_{j \notin \{y_1, y_2\}} e^{z_j^t - 2\eta' \pi_j^t} + e^{z_{y_1}^t - 2\eta' \pi_{y_1}^t + \eta'} + e^{z_{y_2}^t - 2\eta' \pi_{y_2}^t + \eta'}} \quad (25)$$

$$= \frac{e^{z_i^t}}{\sum_{j \notin \{y_1, y_2\}} e^{z_j^t - \eta'\left(2(\pi_j^t - \pi_i^t) + 1\right)} + e^{z_{y_1}^t - 0} + e^{z_{y_2}^t - 2\eta'(\pi_{y_2}^t - \pi_i^t)}} \quad (26)$$

Similarly, for REINFORCE objective,

$$\pi_{y_1}^t = \frac{e^{z_i^{t+1}}}{\sum_{j=1}^{V} e^{z_j^{t+1}}} = \frac{e^{z_i^t - \eta'(\Delta\pi^t - 1)\pi_i^t}}{\sum_{j \notin \{y_1, y_2\}} e^{z_j^t - \eta'(\Delta\pi^t \cdot \pi_j^t)} + e^{z_{y_1}^t - \eta'(\Delta\pi^t - 1)\pi_{y_1}^t} + e^{z_{y_2}^t - \eta'(\Delta\pi^t - 1)\pi_{y_2}^t}} \quad (27)$$

$$= \frac{e^{z_i^t}}{\sum_{j \notin \{y_1, y_2\}} e^{z_j^t - \eta'\left(\Delta\pi^t(\pi_j^t - \pi_i^t) + \pi_i^t\right)} + e^{z_{y_1}^t - 0} + e^{z_{y_2}^t - \eta'\left((\Delta\pi^t - 1)(\pi_{y_2}^t - \pi_i^t)\right)}} \quad (28)$$

For case $i \notin \{y_1, y_2\}$, we have the following derivation for MLE objective:

$$\pi_i^t = \frac{e^{z_i^{t+1}}}{\sum_{j=1}^{V} e^{z_j^{t+1}}} = \frac{e^{z_i^t - \eta'(2\pi_i^t)}}{\sum_{j \notin \{y_1, y_2\}} e^{z_j^t - 2\eta' \pi_j^t} + e^{z_{y_1}^t - 2\eta' \pi_{y_1}^t + \eta'} + e^{z_{y_2}^t - 2\eta' \pi_{y_2}^t + \eta'}} \quad (29)$$

$$= \frac{e^{z_i^t}}{\sum_{j \notin \{y_1, y_2\}} e^{z_j^t - 2\eta'\left(\pi_j^t - \pi_i^t\right)} + e^{z_{y_1}^t - \eta'(2(\pi_{y_1}^t - \pi_i^t) - 1)} + e^{z_{y_2}^t - \eta'(2(\pi_{y_2}^t - \pi_i^t) - 1)}} \quad (30)$$

Similarly for REINFORCE objective:

$$\pi_i^t = \frac{e^{z_i^{t+1}}}{\sum_{j=1}^{V} e^{z_j^{t+1}}} = \frac{e^{z_i^t - \eta'(\Delta\pi^t \cdot \pi_i^t)}}{\sum_{j \notin \{y_1, y_2\}} e^{z_j^t - \eta'(\Delta\pi^t \cdot \pi_j^t)} + e^{z_{y_1}^t - \eta'(\Delta\pi^t - 1)\pi_{y_1}^t} + e^{z_{y_2}^t - \eta'(\Delta\pi^t - 1)\pi_{y_2}^t}} \quad (31)$$

$$= \frac{e^{z_i^t}}{\sum_{j \notin \{y_1, y_2\}} e^{z_j^t - \eta'\left(\Delta\pi^t(\pi_j^t - \pi_i^t)\right)} + e^{z_{y_1}^t - \eta'(\Delta\pi^t(\pi_{y_1}^t - \pi_i^t) - \pi_{y_1}^t)} + e^{z_{y_2}^t - \eta'(\Delta\pi^t(\pi_{y_2}^t - \pi_i^t) - \pi_{y_2}^t)}}$$

$$(32)$$

$$\square$$

We can now better understand how each $\pi_i$ changes after one update. Specifically, if $\alpha_i > 1$, the corresponding $\pi_i$ increases, and vice versa. To determine the value of $\alpha_i$, we can treat any $\beta_j > 1$ as contributing to the conclusion that $\alpha_i < 1$, while any $\beta_j < 1$ against it. The value of the corresponding $e^{z_j^t}$ and $|\beta_j - 1|$ controls how strong the contribution is.

**Low-likelihood tokens tend to receives less meaningful updates.** Using the log-derivative trick, $\nabla_\theta \pi_y = \pi_y \nabla_\theta \log \pi_y$, we see that REINFORCE inherently applies a conservative update rule: tokens with higher probability receive proportionally larger gradient updates, while low-probability tokens are updated much more weakly. This disproportionate updating can have a detrimental effect in cases where optimal actions may reside in low-probability regions of the distribution, yet REINFORCE fails to provide sufficient reinforcement for these tokens to be effectively explored and learned.

**Assuming $\pi_{y_1}^t > \pi_{y_2}^t$, then $\pi_{y_1}^t$ is guaranteed is increase, i.e., $\alpha_{y_1} > 1$.** To see this, consider the value of $\gamma$ in Case 1 of the REINFORCE objective. For any $j \notin y_1, y_2$, we have $\gamma_j < 1$ because

$$\Delta\pi^t(\pi_j^t - \pi_i^t) + \pi_i^t = \Delta\pi^t \pi_j^t + (1 - \Delta\pi^t)\pi_i^t > 0. \quad (33)$$

Moreover, since $\gamma_{y_1} = 1$ while $\gamma_{y_2} < 1$, the probability of the highest-likelihood action $y_1$ will increase. In addition, because the update step size depends on the probability of the sampled action, the smaller $\pi_{y_1}^t$ is, the more strongly $\pi_{y_2}^t$ will decrease, leading the model to gradually concentrate its mass on the dominant mode $\pi_{y_1}^t$. However, this is not the case for the maximum likelihood objective; later, as we show in section 2, the maximum likelihood objective will tend to increase the lower probability $\pi_{y_2}^t$, avoiding model collapse.

### D.1 EMPIRICAL VERIFICATION WITH A SIMPLE BANDIT PROBLEM.

To empirically verify the analysis above, we analyze using a simple logistic regression task following Ren & Sutherland (2025). We set the vocabulary size $V = 50$, with parameter dimension $d = 5$, a learning rate of $\eta = 0.5$, and a randomly generated input vector $\boldsymbol{x}$. We compared the update effect of MLE and REINFORCE objectives in 2 scenarios: when the initial distribution $\pi_{\theta^0}$ is relatively uniform and when it's highly peaked around certain dimensions.

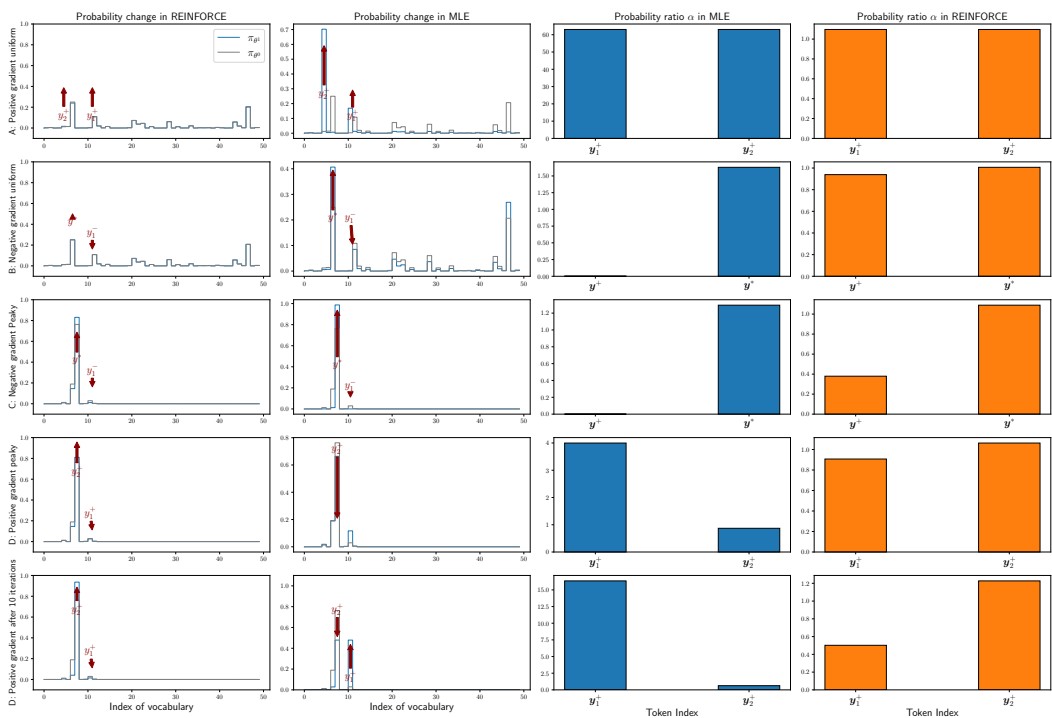

Figure 9: Experimental verification and comparison between REINFORCE and MLE objectives using a simple contextual bandit problem.

**Uniform case:** we consider negative and positive gradient effect, in the negative gradient case, we consider negative label $\boldsymbol{y}^- = 11$ and investigate the "indirect" push-up effect on $\boldsymbol{y}_2 = 5$. In the positive gradient scenario, we use 2 positive actions $\boldsymbol{y}_1 = 11, \boldsymbol{y}_2 = 5$ for the SGD update. As demonstrated in the first 2 rows in Fig. 9, both REINFORCE and MLE behave similarly in both negative and positive gradient scenarios, where both $\boldsymbol{y}_1, \boldsymbol{y}_2$ are increased. However, MLE aggressively imposes a large update, where the probability is approximately 60 times larger on these two tokens after the update, as measured by the probability ratio $\alpha$. Similarly, a negative gradient tends to increase the highest-likelihood token $\boldsymbol{y}^*$, while decreasing other actions, including the positive action $\boldsymbol{y}_2$. This can be explained by REINFORCE by imposing a probability step-size, resulting in a more conservative update, where the models less deviate from their previous step compared to the MLE objective.

**Peaky case:** In LLM regime, the LMs usually initialize from a pre-trained checkpoint, where the model exhibits a non-uniform distribution. We consider the scenario where the models concentrate on a few dimensions in action distribution. Similarly, negative label $\boldsymbol{y}^- = 11$ and investigate the

"indirect" push-up effect on $\boldsymbol{y}_2 = 8$. In the positive gradient scenario, we use 2 positive actions $\boldsymbol{y}_1 = 11, \boldsymbol{y}_2 = 8$ for the SGD update. As shown in the last 3 rows in Fig. 9, when the distribution is peaky, the negative gradient exacerbates the effect of increasing the highest probability token while decreasing other actions, regardless of correctness Ren & Sutherland (2025). Interestingly, in a positive gradient scenario, between the 2 positive actions, MLE tends to push up the actions with the lowest likelihood (as measured by $\alpha$). In the last row, we observe that after 10 iterations of training, MLE can converge to a new distribution with equal probabilities between 2 actions. However, REINFORCE shows favor for the actions with the highest probability; this effect will exacerbate after multiple updates, where it collapses the highest probability $\pi(\boldsymbol{y}_1 = 8)$. This suggests that offline learning can still provide benefits compared to REINFORCE in avoiding model collapse.

# E  ADDITIONAL RESULTS ON DIFFERENT MODELS

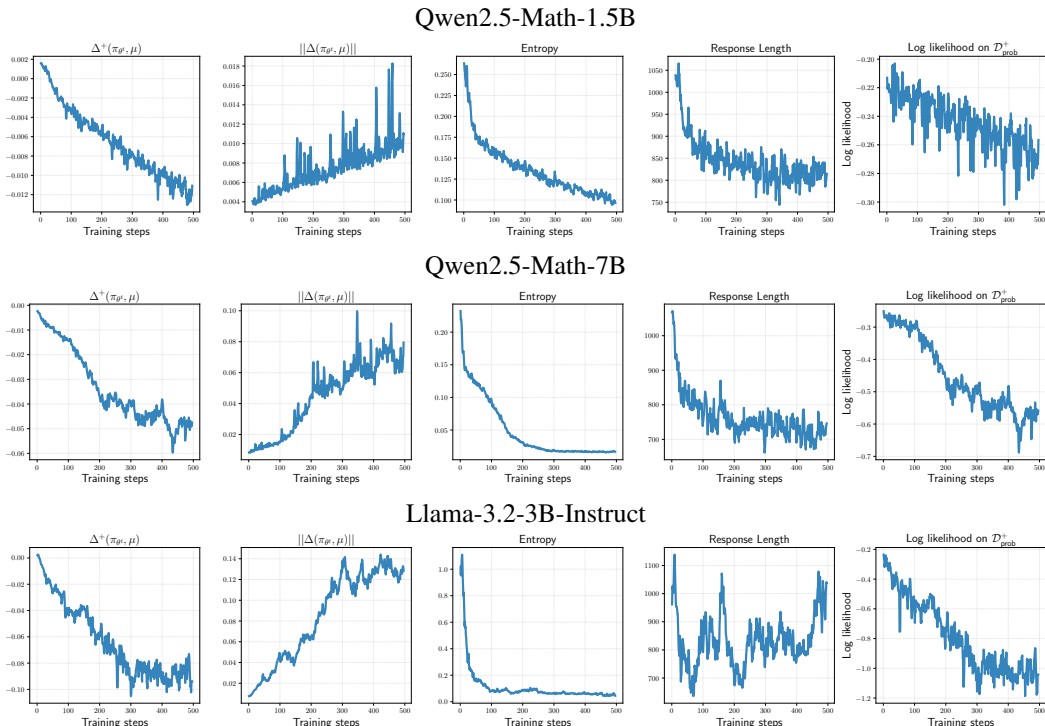

Figure 10: Learning dynamics of RLVR model with different models.

**Consistent learning dynamics across different models.** In this section, we provide additional results that our findings hold across different model families and sizes. Specifically, we conduct GRPO experiments on a larger model, Qwen2.5-Math-7B, as well as a different family, Llama-3.2-3B-Instruct. As shown in Fig. 10, our analysis consistently holds across these settings.

**Larger models tend to suffer more from *negative interference*.** Interestingly, we find that the negative interference phenomenon becomes more severe when scaling up to larger models such as Qwen2.5-Math-7B. This also helps explain why, under RLVR training, the Pass@$k$ performance of Qwen2.5-Math-7B with large sampling budgets $k$ can even underperform than the smaller Qwen2.5-Math-1.5B after GRPO.

**Detailed Pass@$k$ performance.** We present detailed Pass@$k$ curves for Qwen2.5-Math-1.5B and Qwen2.5-Math-7B of different methods below:

Qwen2.5-Math-1.5B

Qwen2.5-Math-7B

Llama-3.2-3B-Instruct

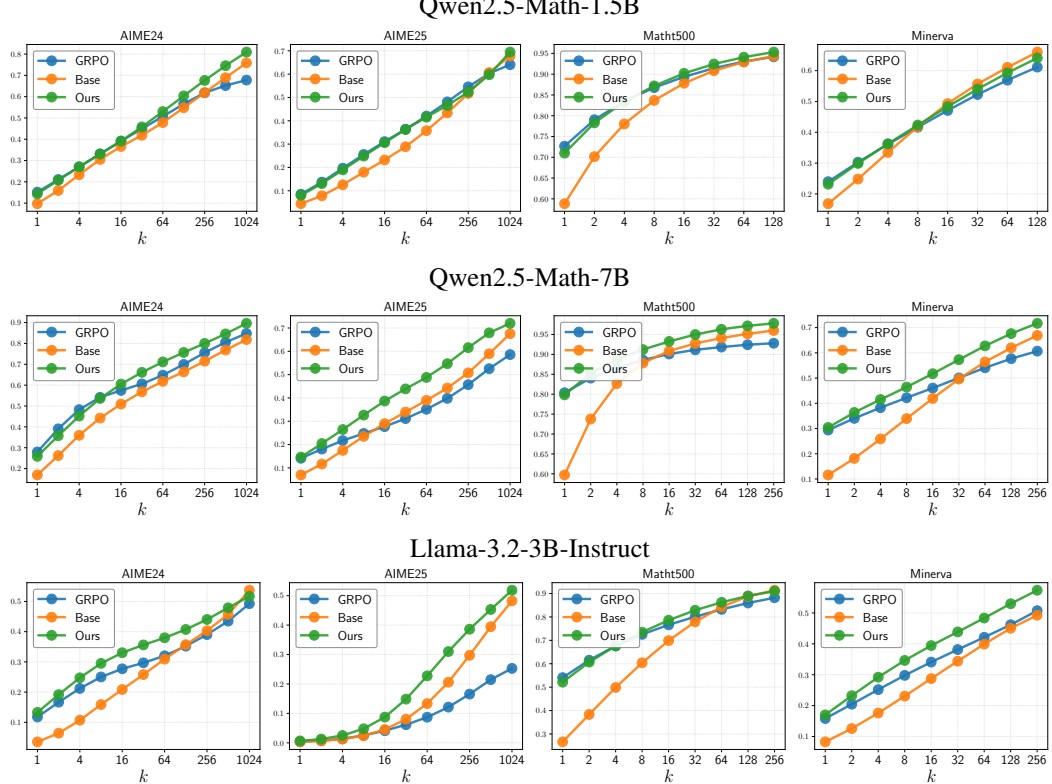

Figure 11: Pass@$k$ curves for base model, RLVR model and SELF with different models.

## F    PERFORMANCE DEGRADATION IN W-REINFORCE

We also experiment with recent work **W-REINFORCE** Zhu et al. (2025), which upweights the negative gradient signal to encourage diversity. As shown in Fig. 2, we find that **W-REINFORCE** suffers significantly from the catastrophic forgetting issue, where the average accuracy decreases even before 300 steps. To understand this collapsing phenomenon arises intrinsically from the al-

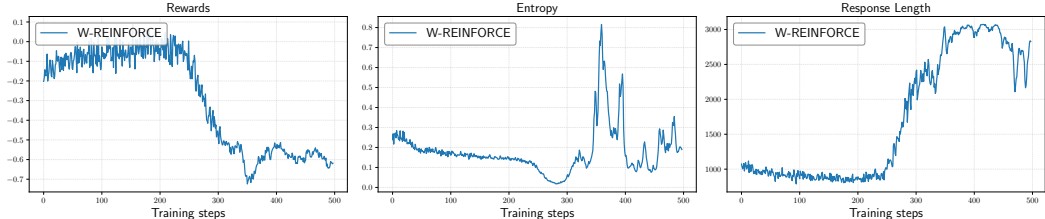

Figure 12: Learning dynamics of **W-REINFORCE**, we observe that **W-REINFORCE** suffer from training instability under prolong training.

gorithms or from numerical precision issues in RLVR training (Qi et al., 2025). We also conduct Float16 mixed-precision for **W-REINFORCE** in Fig. 13. We found that W-REINFORCE still results in a collapsing phenomenon, and even faster than bf16 after a small number of training steps. As a result, this collapsing phenomenon in **W-REINFORCE** is not from numerical precision issues but may arise from the algorithm perspective. We provide the following reasons:

- **High variance issue in W-REINFORCE.** Different from GRPO and other algorithms, W-REINFORCE does not employ any baseline function, which results in a much higher variance policy gradient estimation.

- **Negative gradients in W-REINFORCE can increase the probability of void tokens.**
  W-REINFORCE focuses on learning from negative samples, negative gradients, which can
  indeed increase the exploration ability of LLMs by redistributing reduced probability mass
  to other tokens. However, under prolonged training, this can result in improper increases
  in tail probabilities, leading the model to generate non-meaningful tokens Li et al. (2025b).

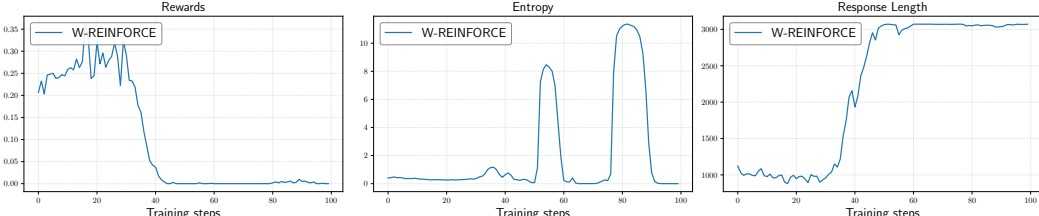

Figure 13: Learning dynamics of **W-REINFORCE** with FP16 Mixed Precision, training still collapses after 100 steps.

## G  COMPUTATIONAL COST OF SELF

Our method require an additional greedy generation to filter highly solvable problems. To analyze
the computational cost of SELF and GRPO, we utilize per-iteration time of each update step with 3
factors: total generation time $T_{\text{gen}}$, backward time $T_{\text{backward}}$, and forward time $T_{\text{forward}}$. We provide
our results in Fig. 14 We find that SELF incurs an additional cost in generation time compared to

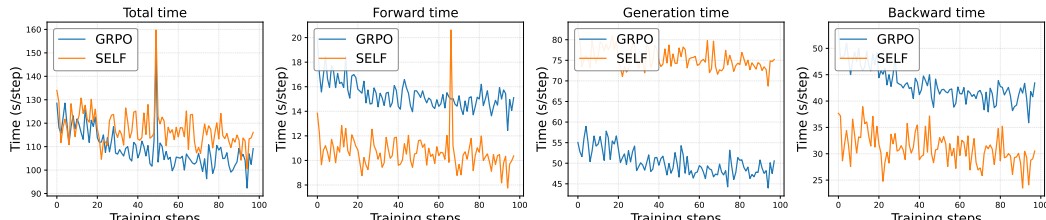

Figure 14: Computational results for training `Qwen2.5-Math-1.5B` for 100 steps.

GRPO. However, due to the filtering mechanism, both the forward and backward passes are faster,
which compensates for the extra generation time and results in a negligible overall computational
cost relative to GRPO.

## H  PERPLEXITY DETAIL ANALYSIS

In this section, we provide a detailed perplexity analysis. Across all benchmarks, we randomly
sample 16 reasoning traces for each problem in the test dataset to calculate perplexity.

To collect problems that increased after RLVR $\mathcal{D}^{\uparrow}$, we estimate the average accuracy for each prob-
lem and sort the problem indices according to average accuracy changes between the base policy
and RLVR policy:

$$\Delta r(\boldsymbol{x}) = \mathbb{E}_{\boldsymbol{y} \sim \pi_{\theta}(\boldsymbol{y}|\boldsymbol{x})}\left[r(\boldsymbol{x}, \boldsymbol{y})\right] - \mathbb{E}_{\boldsymbol{y} \sim \pi_{b}(\boldsymbol{y}|\boldsymbol{x})}\left[r(\boldsymbol{x}, \boldsymbol{y})\right] \tag{34}$$

For AIME 2024/2025, we sample the top-3 problems that increased the most for $\mathcal{D}^{\uparrow}$ and top-3
problems decreased for $\mathcal{D}^{-}$, as these benchmarks only consist of 30 problems. For Math500 and
Minerva, we sample top-64 problems for both $\mathcal{D}^{+}$ and $\mathcal{D}^{\downarrow}$. We present out results in Fig. 15.

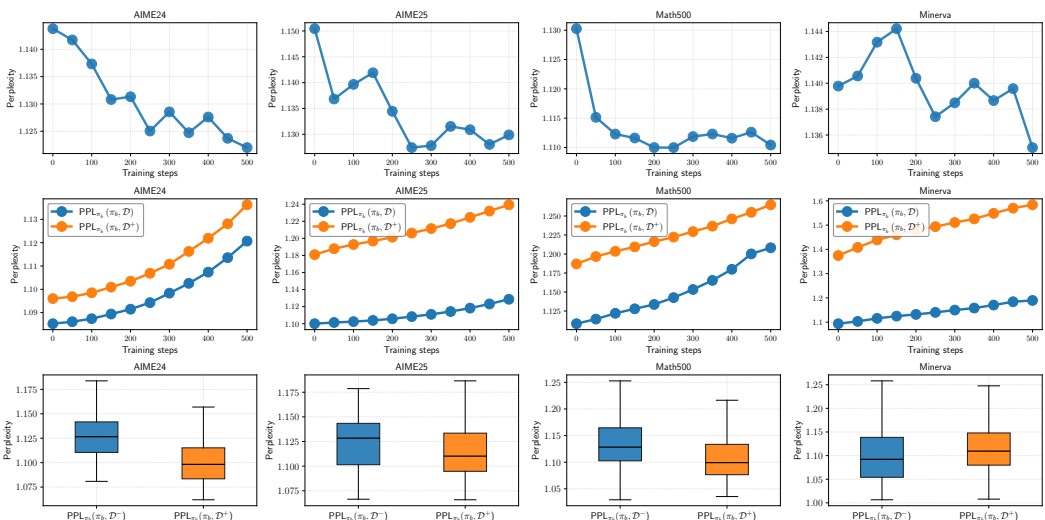

Figure 15: Dynamics of Perplexity during RLVR. The first row shows that RLVR increasingly generates low-perplexity responses under the base model. The second row demonstrates that the initial base solutions, regardless of correctness, exhibit increasing perplexity; the LMs are less likely to generate initial, diverse solutions. The final row shows that for problems where the correct solution already has a high probability to generate under base model, RLVR will incentivize, improving these problems, while problems with low success rate exhibits performance degradation.

# I ADDITIONAL DETAILS IN REDUCING PREVIOUSLY LEARNED BEHAVIORS IN RLVR.

Similar to Shao et al. (2025), we define *code reasoning* are responses that contain the string `python`. For each prompt $\boldsymbol{x}$, the LMs policy can generate either *code reasoning traces* or *lang reasoning traces*. We can define a mixture of distribuion between these two reasoning traces:

$$\pi_b(\boldsymbol{y}|\boldsymbol{x}) = \alpha(\boldsymbol{x})\pi_{\text{code}}(\boldsymbol{y}|\boldsymbol{x}) + (1 - \alpha(\boldsymbol{x}))\pi_{\text{lang}}(\boldsymbol{y}|\boldsymbol{x}) \tag{35}$$

where $\alpha(\boldsymbol{x})$ is the mixture coefficient that depends on the prompt. For each prompt $\boldsymbol{x}$, we sample $k = 144$ responses to estimate the average accuracy of the mixture distributions $\pi_{\text{code}}$ and $\pi_{\text{lang}}$, denoted as $\rho_{\text{code}}(\boldsymbol{x})$ and $\rho_{\text{lang}}(\boldsymbol{x})$. We then define their average accuracies on the test set $\mathcal{D}_{\text{test}}$ as:

$$\mathbb{E}_{\boldsymbol{x}\sim\mathcal{D}_{\text{test}}}\left[\rho_{\text{code}}(\boldsymbol{x})\right], \quad \mathbb{E}_{\boldsymbol{x}\sim\mathcal{D}_{\text{test}}}\left[\rho_{\text{lang}}(\boldsymbol{x})\right]. \tag{36}$$

We report results for each reasoning behavior in the `Qwen2.5-Math` models across all benchmarks below: We observe that the models eventually switch almost entirely to *lang reasoning traces*. We hypothesize that this shift explains the improvements on AIME24/25 and Math500, where *lang reasoning traces* is more effective initially. However, the loss of *code reasoning traces* reduces performance on Minerva, where the base model originally benefited from producing *code reasoning traces*.

# J NEGATIVE INTERFERENCE AND THE EFFECTIVENESS OF SELF IN NON-MATHEMATICAL TASKS.

To asses the existence of negative interference phenomenon and the effectiveness of SELF in other domains, we utilize ReasoningGym Stojanovski et al. (2025), a library of reasoning environments

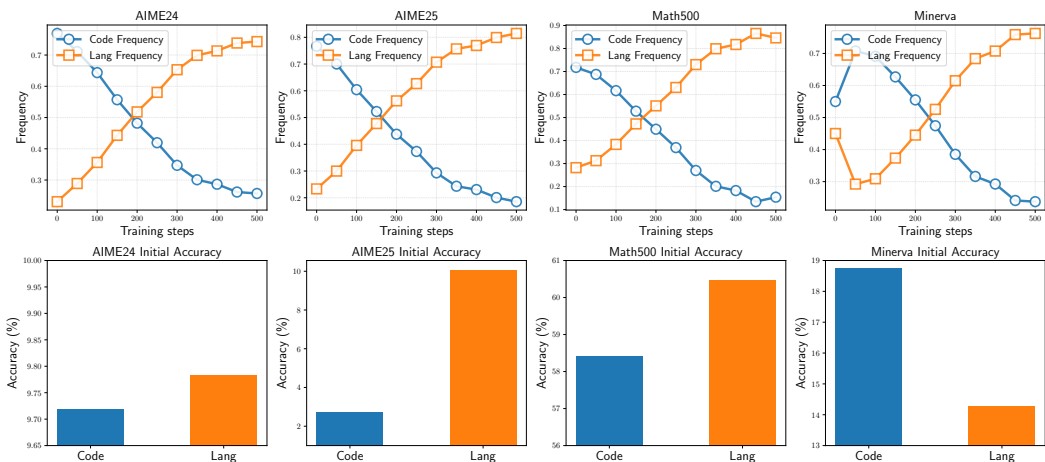

Figure 16: Reasoning strategy switching from `code` to `lang` throughout training process. While AIME24/25 and Math500 exhibits initial better accuracy in `lang` than `code`, Minerva shows better performance in *code reasoning traces*.

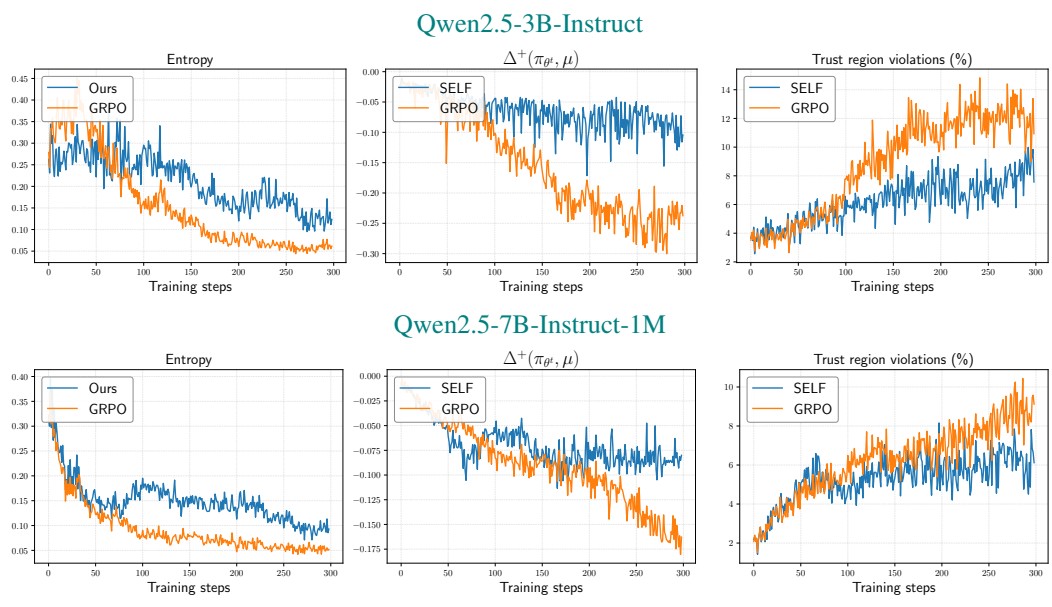

Figure 17: Learning dynamics of SELF and GRPO on ReasoningGym.

for reinforcement learning with verifiable rewards spanning multiple domains such as algebra, arithmetic, cognition, graph theory, logic, etc.

We collect $20k$ training examples spanning across 104 tasks and $2k$ samples for evaluation. We utilize 2 models for this study: Qwen2.5-3B-Instruct and Qwen2.5-7B-Instruct-1M, we measure Pass@$k$ metrics for SELF, GRPO and Base models, following the setups of previous works (Liu et al., 2025a; Stojanovski et al., 2025). We provide our results in Tab. 4 and Fig. 17, which can be summarized below:

- **Negative interference still exists in non-math tasks**. We observed that the increasing negative interference phenomenon still exhibits on non-math tasks. This demonstrates that the negative interference issue is not unique to mathematical domains but also arise in broader reasoning tasks.

Table 4: Evaluation results on ReasoningGym benchmark (%). The best performance under each Pass@$k$ is marked with **boldface**.

| Model | Method | Pass@1 | Pass@4 | Pass@16 | Pass@64 | Pass@128 |
|---|---|---|---|---|---|---|
| | Base model | 16.37 | 26.26 | 35.81 | 43.62 | 46.88 |
| Qwen2.5-3B-Instruct | GRPO | **34.76** | 40.04 | 44.6 | 48.69 | 50.69 |
| | SELF | 31.96 | **41.76** | **48.79** | **53.56** | **55.43** |
| | Base model | 26.85 | 39.16 | 48.2 | 54.89 | 57.75 |
| Qwen2.5–7B-Instruct-1M | GRPO | **42.94** | 49.33 | 54.33 | 59.23 | 61.56 |
| | SELF | 40.67 | **50.98** | **57.74** | **62.34** | **64.3** |

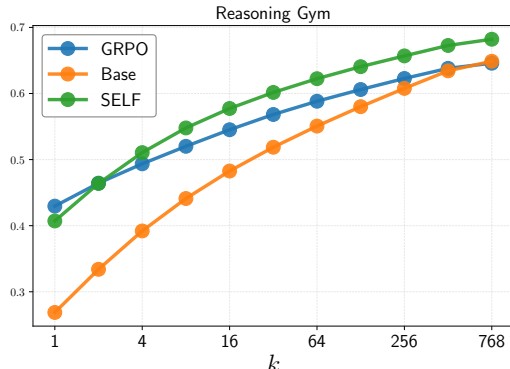

Figure 18: Pass@$k$ curves for Reasoning Gym with Qwen2.5-7B-Instruct-1M.

- **SELF significantly mitigates negative interference and enhances coverage on non-math tasks.** Furthermore, we observe that SELF can effectively mitigate the negative interference phenomenon on non-math tasks, demonstrating that SELF can still generalize to other domains.

- **SELF displays scalability with model size.** We also observed that SELF displays stronger benefits with larger models (as shown in Tab 4 and Fig 18). In particular, the performance gap between SELF and both Base and GRPO grows as we scale from 3B to 7B models, suggesting that larger models can leverage SELF's exploration-guided filtering more effectively.

Our experiments indicate that negative interference is not confined to math reasoning tasks, and demonstrate the effectiveness of SELF in robustly mitigating negative interference and improving solution coverage across a diverse set of non-math reasoning domains, and the effectiveness of SELF scales with model size. These results collectively support our hypothesis that strategies addressing coverage shrinkage in math domains naturally generalize to more general reasoning tasks.

## K ON THE EFFECTIVENESS OF GREEDY FILTERING

In this section, we evaluate the effectiveness of the greedy filtering mechanism in identifying low-likelihood problems. Intuitively, greedy responses provide a deterministic approximation of the highest likelihood solutions. When the greedy response fails on a given problem, it indicates that the model assigns low probability to correct solution paths under its most confident greedy decoding. To support this, we include a boxplot comparing the base model's accuracy on filtered versus non-filtered problems in Fig. 19. We observe a strong correlation between greedy-response failure and problems with low average accuracy, indicating that greedy decoding is an effective signal for identifying low-likelihood or inherently difficult problems with respect to the different models.

**The effectiveness of different decoding strategies.** Intuitively, our findings suggest that problems which are already highly solvable (i.e., those with high average accuracy) under the current model

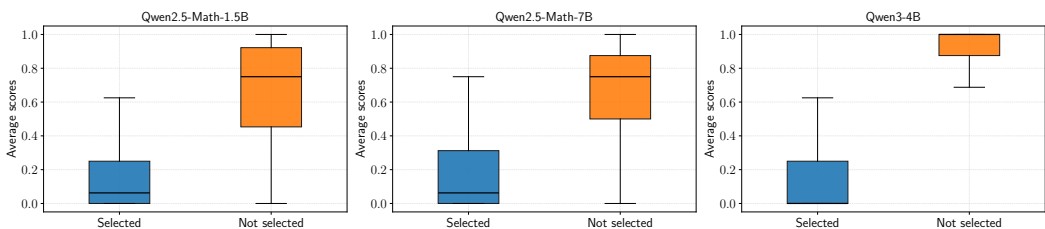

Figure 19: Average accuracy across filtered and non-filtered problems using greedy decoding.

should be filtered. To quantify the effectiveness of different decoding strategies, we classify a problem as highly solvable if its average accuracy satisfies $\mathbb{E}_{y \sim \pi}[r(y)] \geq 0.6$. We estimate this average accuracy by averaging over 16 responses generated with a sampling temperature of $\tau = 0.6$, top-p $= 0.95$, following prior work Yue et al. (2025). To evaluate the alignment between the filtering labels and the predictions of highly solvable problems, we report the F1 score, which accounts for mislabeling errors of the filtering mechanism with recall metric. We further investigate different decoding strategies by varying the sampling parameters: $\tau = \{0.0, 0.6, 0.8, 1.0\}$, top-$p = 0.95$; we provide our results in Tab. 5:

Table 5: F1 score of different decoding strategies across Qwen models.

| Models | Greedy | $\tau = 0.6$ | $\tau = 0.8$ | $\tau = 1.0$ |
|---|---|---|---|---|
| Qwen2.5-Math-1.5B | **74.85** | 69.13 | 74.02 | 60.29 |
| Qwen2.5-Math-7B | **79.06** | 74.5 | 71.92 | 69.1 |
| Qwen3-4B | 91.72 | 92.51 | 90.96 | **92.56** |

We observed that greedy decoding is generally effective and comparable to other decoding strategies. Interestingly, for "thinking" models such as Qwen3, high-accuracy problems often coincide with cases where greedy decoding fails. Furthermore, when greedy decoding fails and tend to produces repetitive tokens, Qwen3 typically achieves low accuracy on those problems. Taken together, these results suggest that greedy decoding can serve as a reliable indicator for identifying low-likelihood solutions.

## L    WHY SELF EXHIBITS DOESN'T OVER-REGULARIZE HIGH-CONFIDENCE PROBLEMS

On-policy algorithms inherently favor high-probability problems, as demonstrated in our analysis (Section 5), while low-likelihood problems receive disproportionately fewer updates. SELF mitigates this imbalance by deliberately emphasizing low-likelihood problems. This emphasis does not over-regularize high-confidence problems because gradients on low-likelihood problems are small in magnitude. Low-likelihood trajectories often produce weaker gradients due to their smaller probability of generating correct solutions; thus, the gradients tend to be weak in magnitude. As a result, even if these problems are sampled more frequently, their updates exert limited cross-problem influence and therefore have minimal impact on high-confidence problems. This trend is clearly shown in Fig. 8, where the per-step relative update strength in SELF is consistently much smaller than in GRPO. We further report the gradient norms of SELF and GRPO over the entire training process in Fig. 20, confirming that SELF operates with substantially smaller gradient magnitudes than GRPO.

## M    USAGE OF LARGE LANGUAGE MODELS (LLMS)

We used ChatGPT solely for revising the writing of the paper. Note that revision here strictly means enhancing the clarity and readability of the text (e.g., fixing typos or constructing latex tables), and not for any other purposes.

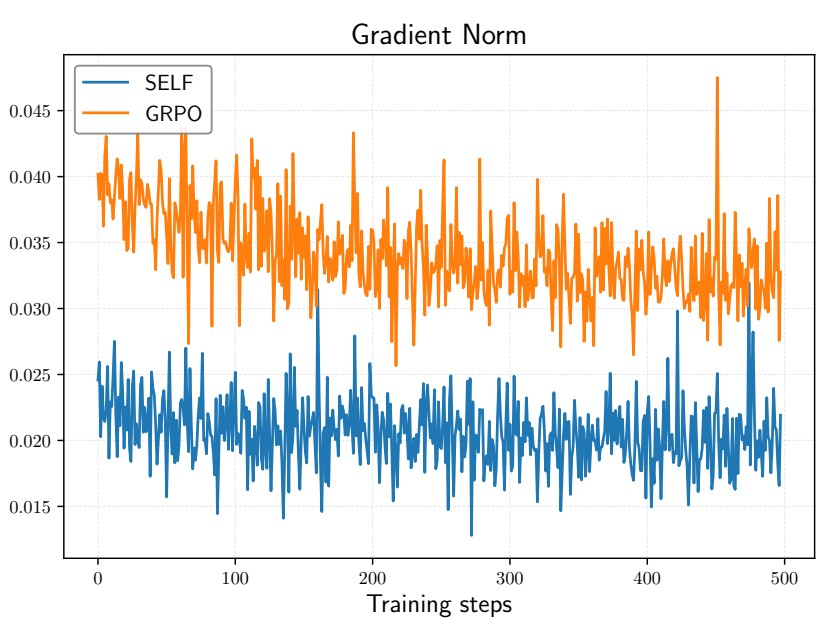

Figure 20: Gradient magnitude of SELF and GRPO throughout training process.

Table 6: Greedy Response Failure Analysis

| Problem String | There are 500 machines, each using 6 parts of the same type. These parts must be replaced by the end of the week if they fail. Out of all new parts, 10% fail by the end of the first week, 30% fail by the end of the second week, and the remaining 60% fail by the end of the third week. No part lasts beyond the third week. How many new parts need to be replaced by the end of the third week? |
|---|---|
| **Average Accuracy** | 6.25% |
| **Greedy Response** | `<think>` Okay, let's try to figure out this problem. So, there are 500 machines, each using 6 parts. These parts need to be replaced by the end of the week if they fail. The failure rates are given for each week: 10% fail by the end of the first week, 30% by the end of the second week, and 60% by the end of the third week. No part lasts beyond the third week. The question is asking how many new parts need to be replaced by the end of the third week. |
| | First, let me make sure I understand the problem correctly. Each machine uses 6 parts. So, in total, how many parts are there initially? Well, 500 machines times 6 parts each would be 500*6 = 3000 parts. But these parts are being used, and they can fail in different weeks. The problem is about how many parts need to be replaced by the end of the third week. |
| | But wait, the parts are being replaced if they fail. So, each part that fails by the end of a week needs to be replaced. However, the problem says that parts must be replaced by the end of the week if they fail. So, maybe the replacement happens at the end of each week? But the question is about the total number of parts that need to be replaced by the end of the third week. So, that would include all parts that failed in the first week, second week, and third week. But maybe there's something else here? |
| | Wait, but the problem says "these parts must be replaced by the end of the week if they fail." So, if a part fails during the week, it needs to be replaced by the end of the week. So, perhaps each week, parts that have failed by the end of that week need to be replaced. But the question is asking for the total number of parts that need to be replaced by the end of the third week. So, that would be the sum of parts that failed in week 1, week 2, and week 3. But maybe there's a catch here. Let me think again. |
| | Alternatively, maybe the parts are being used over the three weeks, and each part can fail in any of the three weeks. But since no part lasts beyond the third week, all parts will have failed by the end of the third week. Therefore, all parts will need to be replaced by the end of the third week. But that can't be right because the problem is giving different failure rates each week. Wait, but maybe the parts that failed in week 1 are already replaced, so they don't need to be replaced again? Or is the replacement happening at the end of each week? |
| | Wait, the problem says "these parts must be replaced by the end of the week if they fail." So, if a part fails during the week, it needs to be replaced by the end of the week. So, for example, if a part fails in week 1, it needs to be replaced by the end of week 1. Similarly, if a part fails in week 2, it needs to be replaced by the end of week 2, and same for week 3. Therefore, by the end of week 3, all parts that have failed in any of the three weeks would have been replaced. However, the question is asking how many new parts need to be replaced by the end of the third week. So, does that mean that we need to count all parts that have failed in any of the three weeks, and thus need to be replaced? But then, since all parts will have failed by the end of week 3, that would be all 3000 parts. But that seems contradictory to the given failure rates. Wait, but maybe the failure rates are not per part, but per the parts that are in use? ... |

Table 7: Greedy Response Failure Analysis

| Problem String | There are 500 machines, each using 6 parts of the same type. These parts must be replaced by the end of the week if they fail. Out of all new parts, 10% fail by the end of the first week, 30% fail by the end of the second week, and the remaining 60% fail by the end of the third week. No part lasts beyond the third week. How many new parts need to be replaced by the end of the third week? |
| --- | --- |
| **Average Accuracy** | 0.0% |
| **Greedy Response** | `<think>` Okay, so I need to figure out the cosine of the angle between the two longer edges of a triangular prism made from these six sticks ... 
 Wait, that seems conflicting. Wait, maybe I need to check again. 
 Wait, a triangular prism has 9 edges. But the problem says we have six sticks. So maybe the problem is not using all edges? Wait, maybe the problem is that the prism is formed by using the sticks as edges. But if there are six sticks, then maybe the prism is not a standard one? Wait, maybe the problem is that the prism is formed by using the sticks as edges, but since there are six sticks, maybe it's a different configuration? Wait, maybe the problem is that the prism is formed by using the sticks as edges, but since a triangular prism has 9 edges, but here we have only six sticks. Therefore, maybe the problem is not referring to a standard triangular prism? Or maybe the problem is referring to a triangular prism where some edges are not all distinct? 
 ... 
 Wait, maybe the problem is referring to a triangular prism where the two triangular bases are not connected by three edges, but instead, maybe the prism is formed by using the sticks as edges, but with some edges being shared? Wait, maybe the problem is that the prism is formed by using the sticks as edges, but the two triangular bases are each made of three sticks, and the three rectangles are each made of two sticks? But that would be 3 + 3 + 3*2 = 9 sticks. But the problem says six sticks. Therefore, maybe the problem is not referring to a triangular prism in the standard way. Maybe it's a different kind of prism? Or maybe the problem is referring to a triangular prism where some edges are not all distinct? Wait, maybe the problem is that the prism is formed by using the sticks as edges, but the two triangular bases are each made of three sticks, and the three rectangles are each made of one stick? But that would be 3 + 3 + 3 = 9 sticks again. Hmm. 
 Alternatively, maybe the problem is referring to a triangular prism where the edges are not all distinct. Wait, maybe the prism is formed by using the sticks as edges, but some edges are shared between the two bases? Wait, but in a prism, the two triangular bases are connected by three edges. So, if we have two triangular bases, each with three edges, and three connecting edges, that's 9 edges. But the problem says six sticks. Therefore, maybe the problem is not referring to a triangular prism with all edges, but maybe a different structure? Maybe the problem is referring to a triangular prism where the two triangular bases are each made of three sticks, and the three rectangles are made of one stick each? But that would be 3 + 3 + 3 = 9 sticks again. Wait, this is confusing. 
 Alternatively, maybe the problem is referring to a triangular prism where the edges are not all distinct. For example, maybe the prism is formed by using the sticks as edges, but some edges are repeated? But that seems odd. Wait, maybe the problem is referring to a triangular prism where the two triangular bases are each made of three sticks, and the three rectangles are made of two sticks each? But that would be 3 + 3 + 3*2 = 9 sticks again. Wait, maybe the problem is not referring to a triangular prism in the standard way. Maybe it's a different kind of prism? Maybe a triangular prism with some edges combined? ... |