# OpenReview forum: "The Reasoning Boundary Paradox: How Reinforcement Learning Constrains Language Models"
_ICLR.cc/2026/Conference — Submitted to ICLR 2026_

### Official Review · Reviewer_jihE · 2025-10-24

**Soundness:** 1
**Presentation:** 4
**Contribution:** 1
**Rating:** 2
**Confidence:** 4

**Summary:**

1. They show the phenomena of negative interference where improving likelihood of correct responses on some prompts reduces likelihood of correct responses on other prompts. This leads to reduction in pass@k performance during RLVR
2. They show that RLVR disproportionately reinforces problems which have high likelihood correct solutions under the base models while suppressing others.
3. Using their insights, they provide a data curation algorithm that helps achieve higher pass@k performance.

**Strengths:**

The problem is quite important and it's important to theoretically understand the root of the issue. The paper tries to look at it from a theoretical and empirical perspective. The paper is presented very well and makes the correct observation that pass@k goes down because pass@1 goes down for certain problems while doing RLVR.

**Weaknesses:**

The analysis done in "Why does RLVR reinforce problems with high-likelihood correct solutions in the base model" from line 342 is erroneous.

1. Low-likelihood tokens provide less meaningful updates:

This is not followed from equation 8 because the gradient's value is a function of the network parameters as well. Also, "less meaningful" is hand wavy. What is exactly meant by less meaningful? Low gradient norm?

2. When there are multiple optimal actions....

This is also not followed from equation 8 because it doesn't take into account the value of the gradient.

**Questions:**

Could the authors comment on the weaknesses and clearly describe any assumptions they've made in section mentioned above?

---

> ### Author Response · Authors · 2025-11-21
>
> We thank the reviewer for their helpful feedback and will respond to the raised questions below.
>
> **Q1: Low-likelihood tokens provide less meaningful updates: This is not followed from equation 8 because the gradient's value is a function of the network parameters as well. Also, "less meaningful" is hand-wavy. What is exactly meant by less meaningful? Low gradient norm?**
>
> **Answer:** We apologize for the ambiguity in the term “less meaningful.” What we mean is that, in on-policy algorithms, the presence of the weighting term $\pi_\theta(y)$ -- which reflects the sampling distribution -- can lead to disproportionately small updates *compared to off-policy approaches*, limiting the reinforcement signal for optimal tokens in low-likelihood regions. In contrast, off-policy approaches such as SFT are not constrained by this weighting, allowing them to produce larger updates even for low-likelihood tokens. Furthermore, these weighting terms also create an additional saddle point, causing zero-probability tokens to remain unupdated. Our analysis focuses on this effect, and our experimental results in the Appendix support this claim, showing that SFT can effectively reinforce low-likelihood target tokens, whereas REINFORCE often fails to do so.
>
> **Q2: When there are multiple optimal actions.... This is also not followed from equation 8 because it doesn't take into account the value of the gradient.**
>
> **Answer:** Thank you for your comment. We'd like to clarify that our analysis fully accounts for the gradient values, and redirect the reviewer to Appendix D. Specifically:
> * Due to limited space, we focus on the probability changes after each update step in the main paper, while both theoretically and empirically analyzing the gradient and showing that actions corresponding to the highest mode are consistently reinforced in the appendix, with their probabilities increasing after each step.
> * Furthermore, our bandit setup illustrates how on-policy learning can create a self-consuming loop, where the policy increasingly concentrates on its own high-probability decisions. This effect becomes even more pronounced in RLVR training, where the analytical expectations are replaced by a finite-budget sampling process of size $k$. Prior work in the self-consuming loop literature [1, 2, 3] has shown that statistical approximation error and function approximation error cause the sampling process to concentrate heavily on the model’s dominant high-probability modes. As a result, low-likelihood (tail) solutions are sampled only rarely, so they contribute zero gradient signal during training. This observation is fully aligned with our analysis in Section 5.
> * Finally, it is very important to note that, in practice, gradient clipping is most often used (or that the gradients of network parameters remain bounded), removing the practical possibility of what is suggested by the reviewer -- i.e., very small probabilities but extremely high gradients.
>
> ---
>     [1] Is Model Collapse Inevitable? Breaking the Curse of Recursion by Accumulating Real and Synthetic Data. COLM 2024
>     [2] Self-Consuming Generative Models with Curated Data Provably Optimize Human Preferences. NeurIPS 2024 Spotlight.
>     [3] The Curse of Recursion: Training on Generated Data Makes Models Forget.

---

> > ### Author Response · Authors · 2025-11-26
> >
> > Dear Reviewer jihE,
> >
> > We appreciate your thoughtful comments.
> >
> > As the discussion period is expected to conclude next week, could you please let us know whether our response addresses your concerns? We are happy to provide any additional clarifications or results to resolve your doubts.
> >
> > Thank you,
> >
> > The Authors

---

### Official Review · Reviewer_Tk7L · 2025-11-01

**Soundness:** 3
**Presentation:** 3
**Contribution:** 3
**Rating:** 6
**Confidence:** 3

**Summary:**

This paper studies why Reinforcement Learning with Verifiable Rewards (RLVR) can sometimes shrink rather than expand reasoning capability in large language models. Through a detailed theoretical and empirical investigation, the authors identify two core failure modes:
(1) Negative interference, where learning to solve some problems reduces the likelihood of solving others, and
(2) Winner-take-all reinforcement, where RLVR disproportionately strengthens problems already solvable by the base model while suppressing low-likelihood ones. They formalize these effects via a per-step influence analysis, empirically validate them on multiple math benchmarks (AIME 2024/25, Math500, Minerva), and propose Selective Examples with Low-likelihood and Forward-KL (SELF), a data-curation algorithm that focuses learning on low-success-rate examples. SELF mitigates coverage shrinkage and improves Pass@k consistency without extra computational cost.

**Strengths:**

1. The per-step influence formulation provides an insightful view of how on-policy RLVR gradients cause cross-problem interference and loss of coverage. This formalism connects RLVR behavior to known interference and plasticity-loss phenomena in RL
2. The proposed method SELF is conceptually simple (i.e., curating low-likelihood examples and using forward-KL regularization) yet effectively improves coverage and diversity (entropy, trust-region violations).
3. The experiments are thorough, spanning three model families and four reasoning benchmarks, showing consistent Pass@k degradation under standard RLVR and improvement with SELF. The correlation analyses between interference metrics and Pass@k drop are particularly strong and interpretable

**Weaknesses:**

1. Additional exploration beyond math reasoning would further strengthen the work. All benchmarks are math-centric, it remains unclear whether the observed interference and SELF’s benefits generalize to code/commonsense reasoning tasks.
2. Missing discussion of potential trade-offs. SELF improves Pass@k but sometimes slightly reduces Pass@1. A deeper analysis of why emphasizing low-likelihood problems doesn’t over-regularize high-confidence ones would strengthen understanding of its limits.

**Questions:**

- Would the negative interference also appears in non-verifiable/non-math RL scenarioes?

- How sensitive is SELF to the accuracy of low-likelihood identification? For example, does mis-labeling easy problems as “low-likelihood” hurt stability?

- Regarding the training stability, was mixed-precision training used (bf16 or fp16)? Given that RLVR often face numerical precision issues, it would be helpful to clarify whether the reported instability of baselines (e.g., GRPO/W-REINFORCE) arises intrinsically from the algorithms or from precision-mode effects. Would GRPO/W-REINFORCE still suffer from training divergence if training in fp16?

---

> ### Author Response · Authors · 2025-11-21
>
> Thank you for your thoughtful review and valuable feedback. Please find our response to your comments below.
>
> **Q1: Additional exploration beyond math reasoning would further strengthen the work. All benchmarks are math-centric; it remains unclear whether the observed interference and SELF’s benefits generalize to code/commonsense reasoning tasks. Would the negative interference also appear in non-verifiable/non-math RL scenarios?**
>
> **Answer:** Thank you for your comments. Our experiments closely follow the setup of prior works [1,2,3], which only focus on mathematical reasoning. We hypothesize that methods addressing coverage shrinkage in mathematical domains—where base models are often overtrained during pre-training, limiting their exploration ability [4]—can be naturally extended to other domains.
>
> Nevertheless, following your suggestions, we also tested the existence of negative interference and the effectiveness of SELF in additional tasks, using ReasoningGym [5], which covers multiple domains such as games, logic puzzles, algorithmic, cognition, etc. We collect 20k training examples across these tasks and 2k samples for evaluation. We utilize 2 models for this study: Qwen2.5-3B-Instruct and Qwen2.5-7B-Instruct-1M; we measure Pass@1 and Pass@128 for SELF, GRPO, and the Base model. Our results are summarized below:
>
> Qwen2.5-3B-Instruct Reasoning Gym result
>
> | Method | Pass@1 | Pass@4 | Pass@16 | Pass@64 | Pass@128 |
> | :--- | :--- | :--- | :--- | :--- | :--- |
>   | Base model | 16.37 | 26.26 | 35.81 | 43.62 | 46.88 |
>  | GRPO | **34.76** | 40.04 | 44.6 | 48.69 | 50.69 |
> | SELF | 31.96 | **41.76** | **48.79** | **53.56** | **55.43** |
>
> Qwen2.5-7B-Instruct-1M Reasoning Gym result
>  Method | Pass@1 | Pass@4 | Pass@16 | Pass@64 | Pass@128 |
> | :--- | :--- | :--- | :--- | :--- | :--- |
> |  Base model | 26.85 | 39.16 | 48.2 | 54.89 | 57.75 |
> |  GRPO | **42.94** | 49.33 | 54.33 | 59.23 | 61.56 |
> | SELF | 40.67 | **50.98** | **57.74** | **62.34** | **64.3** |
>
> - **Negative interference still exists in non-math tasks.** We observed that the increasing negative interference phenomenon still exhibits on non-math tasks. This demonstrates that the negative interference issue is not unique to mathematical domains but also arises in broader reasoning tasks. Comprehensive results for the non-math tasks are presented in Appendix J, Figure 16, in the revised manuscript.
> - **SELF significantly mitigates negative interference and enhances coverage on non-math tasks**. Furthermore, we observe that SELF can effectively mitigate the negative interference phenomenon on non-math tasks, demonstrating that SELF can still generalize to other domains.
> - **SELF displays scalability with model size.** We also observed that SELF displays stronger benefits with larger models. In particular, the performance gap between SELF and both Base and GRPO grows as we scale from 3B to 7B models, suggesting that larger models can leverage SELF’s exploration-guided filtering more effectively.
>
> Overall, our experiments indicate negative interference is not confined to math reasoning tasks, and demonstrate the effectiveness of SELF in mitigating negative interference and improving solution coverage across a diverse set of non-math reasoning problems, and the effectiveness of SELF scales with model size. These results collectively support our hypothesis that strategies addressing coverage shrinkage in math domains naturally generalize to more general reasoning tasks.
>
> **Q2: Missing discussion of potential trade-offs. SELF improves Pass@k but sometimes slightly reduces Pass@1. A deeper analysis of why emphasizing low-likelihood problems doesn’t over-regularize high-confidence ones would strengthen understanding of its limits.**
>
> **Answer:** Thank you for your suggestion. On-policy algorithms *inherently* favor high-probability problems, as demonstrated in our analysis (Section 5), while low-likelihood problems receive disproportionately fewer updates. SELF mitigates this imbalance by deliberately emphasizing low-likelihood problems. This emphasis does not over-regularize high-confidence problems because **gradients on low-likelihood problems are small in magnitude.**
>
> Low-likelihood trajectories often produce weaker gradients due to their smaller probability of generating correct solutions; thus, the gradients tend to be weak in magnitude. As a result, even if these problems are sampled more frequently, their updates exert limited cross-problem influence and therefore have minimal impact on high-confidence problems. This trend is clearly shown in Fig. 8, where the per-step relative update strength in SELF is consistently much smaller than in GRPO. We further report the gradient norms of SELF and GRPO over the entire training process in Fig.18 in our revised manuscript, confirming that SELF operates with substantially smaller gradient magnitudes than GRPO.

---

> > ### Author Response · Authors · 2025-11-21
> >
> > **Q3: How sensitive is SELF to the accuracy of low-likelihood identification? For example, does mis-labeling easy problems as “low-likelihood” hurt stability?**
> >
> > **Answer:** Thank you for your suggestion. Intuitively, based on our findings, we should filter problems that are already highly solvable (i.e., those with high average accuracy) under the current model. We provide the experiments showing the accuracy of this "identification" problem with greedy and other decoding strategies. Specifically, we classify problems as highly solvable if the average accuracy satisfies $\mathbb E_{y\sim\pi}[r(y)] > 0.6$. We report the F1 score measuring the alignment between the filtering labels and the predictions of highly solvable problems under different decoding strategies by changing temperature $\tau=\{0.0, 0.6,0.8,1.0\}, \text{top-}p=0.95$:
> >
> > | Decoding strategies\Model | Qwen2.5-Math-1.5B | Qwen2.5-Math-7B | Qwen3-4B|
> > | -------- | -------- | -------- | - |
> > | Greedy ($\tau=0.0$)     |    **74.85**  |  **79.06**   | 91.72
> > | $\tau=0.6$ |69.13 | 74.5 | 92.5 |
> > |$\tau=0.8$ | 74.02|71.92 | 90.9
> > |$\tau=1.0$ |60.29 | 69.10 | 92.5|
> >
> > We observed that greedy decoding is strongly aligned with identifying low-likelihood problems. Interestingly, for “thinking” models such as Qwen3, high-accuracy problems tend to coincide with cases where greedy decoding fails. Moreover, when greedy decoding fails and produces repetitive tokens, Qwen3 typically achieves low accuracy on those problems. Taken together, these findings indicate that greedy responses can serve as an effective signal for identifying low-likelihood solutions.
> >
> > **Q4: Regarding the training stability, was mixed-precision training used (bf16 or fp16)? Given that RLVR often face numerical precision issues, it would be helpful to clarify whether the reported instability of baselines (e.g., GRPO/W-REINFORCE) arises intrinsically from the algorithms or from precision-mode effects. Would GRPO/W-REINFORCE still suffer from training divergence if training in fp16?**
> >
> > **Answer:** Thank you for your question. In our experiments, we didn't observe the collapsing problem in both SELF and GRPO under BF16 precision, but did observe it in W-REINFORCE. Taking your suggestion, we conducted FP16 mixed precision training with W-REINFORCE using hyperparameters suggested by the authors. We found that W-REINFORCE still results in the collapsing phenomenon, and even faster than bf16 after a small number of training steps. We hypothesize this collapsing problem not from numerical precision issues but from the algorithm perspective, with the following reasons:
> > - **High variance issue in W-REINFORCE.** Different from GRPO and other algorithms, W-REINFORCE does not employ any baseline function, which results in a much higher variance policy gradient estimation.
> > - **Negative gradients in W-REINFORCE can increase the probability of void tokens.** W-REINFORCE focuses learning on negative samples; negative gradients, which can indeed increase the exploration ability of LLMs by redistributing, reducing the probability mass to other tokens. However, this redistribution is *undirected*, meaning that there is no explicit guidance on which tokens should be redistributed. As a result, under prolonged training, this can result in improper increases in tail probabilities (tokens that are non-relevant to context), leading the model to generate non-meaningful tokens [6].
> >
> > ---
> >     [1] The Surprising Effectiveness of Negative Reinforcement in LLM Reasoning. NeurIPS 2025.
> >     [2] DAPO: An Open-Source LLM Reinforcement Learning System at Scale. NeurIPS 2025.
> >     [3] Learning to Reason under Off-Policy Guidance. NeurIPS 2025.
> >     [4] ProRL: Prolonged Reinforcement Learning Expands Reasoning Boundaries in Large Language Models. NeurIPS 2025.
> >     [5] Reasoning Gym: Reasoning Environments for Reinforcement Learning with Verifiable Rewards. NeurIPS 2025 Benchmark Track.
> >     [6] Preserving Diversity in Supervised Fine-Tuning of Large Language Models. ICLR 2025.

---

> ### Author Response · Authors · 2025-11-25
> **SELF can expand reasoning boundary beyond the base model on non-math centric tasks**
>
> To evaluate whether SELF can truly expand the reasoning boundary beyond the base model on non-math tasks, we updated new Reasoning Gym Pass@k results using Qwen2.5‑7B‑Instruct‑1M with a larger sampling budget ($k$ = 768). At a large sampling budget $k$, the Base model begins to surpass GRPO, indicating that GRPO does not really expand the reasoning frontier beyond the base model. In contrast, SELF consistently outperforms both GRPO and the Base model across a broad range of $k$, with a large margin of 4%. Our updated results show that SELF can truly expand the reasoning boundary beyond the base model on tasks that are not primarily math-centric. The results are reported in the table below:
>
> | Methods | Pass@1 | Pass@4 | Pass@16 | Pass@64| Pass@256 | Pass@768|
> | -------- | -------- | -------- | - | -| -| -|
> | Base     |  26.87    | 39.2  | 48.26 | 55.07 | 60.77 | 64.87
> | GRPO |**42.95**  | 49.36 |  54.51 | 58.82 | 62.27| 64.58
> | SELF |  40.71  | **51.05** | **57.73** | **62.24** | **65.69**| **68.21**
>
> We hope these clarifications and our new experimental results (which we added to the final paper in Fig.18) will fully address the reviewer's concerns. We thank the reviewer again for their time and valuable feedback.

---

> > ### Author Response · Authors · 2025-11-26
> >
> > Dear Reviewer Tk7L,
> >
> > We appreciate your constructive and positive feedback.
> >
> > As the discussion concludes next week, could you please let us know whether our response addresses your concerns? We are happy to provide any additional clarifications or results to resolve your doubts.
> >
> > Thank you,
> >
> > The Authors

---

### Official Review · Reviewer_AE2C · 2025-11-09

**Soundness:** 3
**Presentation:** 3
**Contribution:** 2
**Rating:** 6
**Confidence:** 3

**Summary:**

This paper investigates how RLVR affects reasoning boundary of LLMs, measured by pass@k performance. The authors identify two phenomena: 1. negative interference, where learning to solve certain problems reduces likelihood of others, and 2. winner-take-all, where RLVR disproportionately reinforces problems with high-likelihood correct solutions while suppressing low-likelihood ones. The proposed method SELF, which focuses on low-likelihood problems and uses forward KL regularization, proves to be effective in improving LLM reasoning while preserve base model's pass@k performance.

**Strengths:**

1. The per-step influence provides a principled analysis to understand how updates on one problem affect others, which is comprehensive and well supported by the experiments in section 4.2.
2. The proposed method SELF is simple to implement and shows consistent improvements in pass@k across benchmarks while maintaining comparable pass@1 performance.

**Weaknesses:**

1. The main finding of this paper is well studied in many priors works, as also mentioned in the paper, therefore weakening the novelty.
2. The paper does not provide sufficient justification for why filtering based on greedy response failure is an appropriate or effective way to identify low-likelihood problems. Other potential strategies for targeting such problems could be explored. Moreover, the proposed filtering approach may not generalize well to models whose optimal decoding strategy is sampling-based rather than greedy, such as Qwen3 Thinking models.

**Questions:**

1. The proposed SELF objective is intended to train only on problems where the greedy response fails. However, Eq. (9) appears to optimize examples where the greedy answers ( $y^\*$ ) are correct, since the indicator function ( $\mathbf{1}(r(x, y^\*) \in C(x))$ ) selects successful responses. Could the authors clarify this discrepancy?
2. Are the improvements reported in Table 1 statistically significant? For example, are they at least beyond the 2-sigma or 95% confidence level?

---

> ### Author Response · Authors · 2025-11-21
>
> Thank you for your review and positive feedback on our paper. We appreciate your acknowledgment of the clarity and organization of our work.
>
> Please find our response to your comments below.
>
> **Q1: The main finding of this paper is well studied in many priors works, as also mentioned in the paper, therefore weakening the novelty.**
>
> **Answer:** Thank you for your comment and efforts in reviewing our paper. We would like to emphasize that our paper provides significant contributions beyond all prior works. Specifically,
>
> - **Our paper explains the coverage shrinkage in RLVR.** The previous works primarily "observe" the coverage shrinkage phenomenon in RLVR training, but *our work explains why this phenomenon emerges*. Specifically, we provide both theoretical and empirical analysis, which reveals the *negative cross-interference* mechanism and *winner-take-all* phenomenon, and the consequences of the on-policy learning, uncovering the main drivers for coverage shrinkage. Consequently, our work provides a deeper insight into the root causes of coverage shrinkage and a foundation for future mitigation methods.
> - **Our paper provides a strong connection to cross-interference in traditional RL to coverage shrinkage problem in RLVR.** We propose 2 lightweight, qualitative metrics that characterize how model's behavior evolves throughout RLVR training. These metrics enable us to establish a strong connection between negative interference and plasticity loss in traditional RL [1, 2, 3] and the coverage-shrinkage phenomenon in RLVR [4] (Sec. 4.2). As acknowledged by Rev. Tk7L, this connection is a meaningful contribution, because it explains that on-policy learning and the growing of the negative interference cause the LLMs to lose reasoning abilities to solve other problems (Sec. 5). Importantly, this connection enables us to borrow mitigation strategies from *well-known works* in traditional RL to RLVR. For instance, regularization methods as in [5, 6] can be adapted to RLVR to reduce harmful interference, ensuring that learning on a problem $x$ does not disproportionately affect performance on other problems, or motivate a deeper investigation of off-policy and hybrid objectives as potential ways to circumvent this issue. We leave these for future works.
>
> - **Our paper provides a novel, data-efficient method to solve the coverage shrinkage problem.** To the best of our knowledge, prior work has primarily analyzed the coverage-shrinkage phenomenon without providing consistent or reliable solutions to mitigate it. Our paper is among the first to propose an extremely simple yet highly effective method that directly targets coverage shrinkage while avoiding the training instability commonly observed in RLVR training.
>
> In summary, we believe that these contributions are significant and original, beyond the findings in existing works, and we hope the response has addressed the reviewer's concerns. We thank the reviewer again for the thoughtful feedback.
>
> **Q2. Other potential strategies for targeting such problems could be explored.**
>
> **Answer:** Thank you for your comment. As mentioned above, by providing a connection between negative cross-interference problem in traditional RL and coverage shrinkage in RLVR, this allows mitigation strategies in *well-known problems* to be applied to RLVR. For instance, regularization methods such as those in [5, 6] can be adapted to RLVR to reduce harmful interference, ensuring that learning on one training problem $x$ does not disproportionately affect performance on other problems, or motivate a deeper investigation of off-policy and hybrid objectives as potential ways to circumvent this issue. Nevertheless, these extensions are beyond the scope of our work ( and deserve independent studies), which focuses on explaining coverage shrinkage.
>
> **Q3. The proposed SELF objective is intended to train only on problems where the greedy response fails. However, Eq. (9) appears to optimize examples where the greedy answers $y^*$ are correct, since the indicator function $\left(\mathbb 1\{y\in\mathcal C(x)\}\right)$ selects successful responses. Could the authors clarify this discrepancy?**
>
> **Answer:** Thank you a lot for catching this — this is indeed a typo in the equation. The indicator function in Eq. (9) should select problems where the greedy response fails, not successful ones. In other words, it should be $\mathbb 1\{y\notin\mathcal C(x)\}$. We corrected this in the revised version.
>
> **Q4. Are the improvements reported in Table 1 statistically significant? For example, are they at least beyond the 2-sigma or 95% confidence level?**
> - **Answer:** Thank you for your comment. Our results is reported is take into account of statistical significant average across 3 random training seeds. We updated statistical significane results in the main paper.

---

> > ### Author Response · Authors · 2025-11-21
> >
> > **Q5. The paper does not provide sufficient justification for why filtering based on greedy response failure is an appropriate or effective way to identify low-likelihood problems. Moreover, the proposed filtering approach may not generalize well to models whose optimal decoding strategy is sampling-based rather than greedy, such as Qwen3 Thinking models.**
> >
> > **Answer:** Thank you for your insightful comment. We address the two concerns below.
> > - **Justification for filtering based on greedy-response failure.** Greedy responses provides a deterministic approximation of highest likelihood solutions. When the greedy response fails on a given problem, it indicates that the model assigns low probability to correct solution paths under its most confident greedy decoding. To further support this, we include a boxplot comparing the base model’s accuracy on filtered versus non-filtered problems. Empirically, we observe a strong correlation between greedy-response failure and problems with low average accuracy, indicating that greedy decoding is an effective signal for identifying inherently difficult problems with low-likelihood solutions.
> > - **Generalization to other decoding models.** Intuitively, based on our findings, we should filter problems that are already highly solvable (i.e., those with high average accuracy) under the current model. To quantify the effectiveness of different decoding strategies, we classify problems as highly solvable if the average accuracy satisfies $\mathbb E_{y\sim\pi}[r(y)] > 0.6$. We report the F1 score measuring the alignment between the filtering labels and the predictions of highly solvable problems under different decoding strategies by changing temperature $\tau=\{0.0, 0.6,0.8,1.0\}, \text{top-}p=0.95$:
> >
> >
> > | Decoding strategies\Model | Qwen2.5-Math-1.5B | Qwen2.5-Math-7B| Qwen3-4B|
> > | -------- | -------- | -------- | - |
> > | Greedy ($\tau=0.0$)     |   **74.85**   | **79.06** |    91.724
> > | $\tau=0.6$ |69.13 | 74.5 | 92.51
> > |$\tau=0.8$ | 74.02|71.92 | 90.96
> > |$\tau=1.0$ |60.29 | 69.10 |**92.56**
> >
> > We observed that greedy decoding is strongly aligned with identifying low-likelihood problems. Interestingly, for “thinking” models such as Qwen3, with non-greedy decoding, low-accuracy problems tend to coincide with cases where greedy decoding fails. Moreover, when greedy decoding fails and produces repetitive tokens, Qwen3 typically achieves low accuracy on those problems. Taken together, these findings indicate that greedy responses can serve as an effective signal for identifying low-likelihood solutions, irrespective of decoding strategy.
> >
> > ## References
> > ----
> >     Ray Interference: a Source of Plateaus in Deep Reinforcement Learning. arXiv:1904.11455.
> >     Measuring and Mitigating Interference in Reinforcement Learning. arXiv:2307.04887.
> >     Towards a practical measure of interference for reinforcement learning. arXiv:2007.03807.
> >     Does Reinforcement Learning Really Incentivize Reasoning Capacity in LLMs Beyond the Base Model? NeurIPS 2025 Oral.
> >     No Representation, No Trust: Connecting Representation, Collapse, and Trust Issues in PPO. NeurIPS 2024.
> >     Improving Deep Reinforcement Learning by Reducing the Chain Effect of Value and Policy Churn. NeurIPS 2024.

---

> > ### Author Response · Authors · 2025-11-26
> >
> > Dear Reviewer AE2C,
> >
> > We sincerely appreciate your constructive review.
> >
> > As the discussion period is expected to conclude next week, could you please let us know whether our response addresses your concerns? We are happy to answer any further questions or provide necessary clarifications or results to resolve your doubts.
> >
> > Thank you,
> >
> > The Authors

---

### Author Response · Authors · 2025-11-28

`Note: we have updated this final comment in light of the rebuttal process change, after our previously posted version and during the discussion period.`

We thank all of the reviewers for their time and their detailed and constructive comments. We are encouraged to see the reviewers align on the following key strengths of our work:

- **Timeliness and Importance Problem.** While previous works primarily "observe" the coverage shrinkage phenomenon in RLVR training (*i.e., RLVR does not expand, and even shrink, the reasoning boundary beyond what the base model already possesses*), our work is the 1st that explains the root causes of this issue, which is timely and important as appreciated by both Reviewers jihE and jihE.
- **Comprehensive and Insightful Analysis.** To uncover the main drivers of coverage shrinkage, we provided both comprehensive theoretical and empirical analysis, including the **negative cross-interference** mechanism and **winner-take-all** phenomenon, the **consequences of the on-policy learning**, and **limitations of existing regularization techniques**. In addition, our paper provides a strong connection between cross-interference in traditional RL and the coverage shrinkage problem in RLVR. Consequently, our work not only provides a deeper insight into the root causes of coverage shrinkage (as acknowledged by Reviewers AE2C and Tk7L), but also lays the groundwork for future mitigation methods by adapting mitigation strategies from classical RL to the RLVR setting (as acknowledged by Reviewer Tk7L).
- **Simple yet effective mitigation approach with strong empirical results.** Our proposed method, SELF, consistently improves coverage and diversity by selectively training only on problems where the greedy response fails, achieving effective results while being extremely simple and data efficient (as appreciated by Reviewers Tk7L and AE2C). Despite being simple, SELF is grounded on strong empirical observations that **both greedy and non-greedy decoding are strongly aligned with identifying low-likelihood problems**.

---

> ### Author Response · Authors · 2025-12-04
>
> We appreciate this positive feedback and will now summarize our responses to the main concerns from the reviewers:
>
> **[Reviewer AE2C] The Novelty of our work.** raises concerns of "weakened novelty". We highlighted our unique contributions: we are the first to provide an explanation on why *coverage shrinkage* emerges during RLVR training through both empirical and theoretical analysis (Sections 4 and 5). We also highlighted our analysis provides a strong connection to existing studies of traditional RL to *coverage shrinkage isuse* [4,5] (as acknowledged by Rev. Tk7L). Furthermore, to the best of our knowledge, we are the first to introduce an extremely simple yet highly effective method that directly targets the coverage shrinkage issue. We believe that these contributions are significant and original, beyond the findings in existing works.
>
> **[Reviewer Tk7L] The existence of negative interference and the effectiveness of SELF in non-math centric tasks.** Reviewer Tk7L raises concern of the existence of negative interference and the effectiveness of SELF in non-math centric tasks. While we clarified that our work closely follow the setup of prior works [1,2,3], we still provided additional experiments on ReasoningGym (Appendix J) -- which spans across various domains such as games, logic puzzles, algorithmic, cognition, etc... We show that negative interference phenomenon still exists in non-math tasks (Fig. 17). We also highlighted that SELF is still extremely effective on other domains and can even *broaden* the reasoning boundary beyond the base model (Fig. 18).
>
> **[Reviewers AE2C and TK7L] The effectiveness of greedy decoding strategies.** Reviewers AE2C and TK7L raise concerns on the effectiveness of greedy decoding strategies in identifying problems with low-likelihood of producing correct solutions. We provide additional empirical analyses showing that problems on which greedy decoding fails tend to exhibit substantially low average accuracy and conversely, problems solved by greedy decoding generally correspond to high-accuracy regions across different models compared to other decoding strategies (Fig. 19 and Tab. 5).
>
> **[Reviewer jihE] Equation 8 does not take into account of the gradient values of network parameters.** Reviewer jihE raises concern that Equation 8 does not take into account the gradient value of network parameters. We clarified that this is an **incorrect characterization of our work** as our analysis does, in fact, take into account the gradient value of network parameters (Appendix D). Moreover, our analysis is consistent and even more pronounced during RLVR training, where the sampling process tends to concentrate on the model’s dominant modes, causing low-likelihood correct solutions to almost never be sampled [6,7,8], contributing zero gradient during training.
>
> In summary, we have exhaustively responded to all concerns raised by the reviewers. We also believe that these concerns involve either clarification or addition of minor analysis/results. All responses are now incorporated in our revised manuscript.
>
>     [1] The Surprising Effectiveness of Negative Reinforcement in LLM Reasoning. NeurIPS 2025.
>     [2] DAPO: An Open-Source LLM Reinforcement Learning System at Scale. NeurIPS 2025.
>     [3] Learning to Reason under Off-Policy Guidance. NeurIPS 2025.
>     [4] Ray Interference: a Source of Plateaus in Deep Reinforcement Learning. arXiv:1904.11455
>     [5] Improving Deep Reinforcement Learning by Reducing the Chain Effect of Value and Policy Churn. NeurIPS 2024.
>     [6] Is Model Collapse Inevitable? Breaking the Curse of Recursion by Accumulating Real and Synthetic Data. COLM 2024
>     [7] Self-Consuming Generative Models with Curated Data Provably Optimize Human Preferences. NeurIPS 2024 Spotlight.
>     [8] The Curse of Recursion: Training on Generated Data Makes Models Forget. arXiv:2305.17493

---

### Meta-Review · Area_Chair_vKYT · 2026-01-09

**Summary:**

The paper analysis the decrease of pass@k phenomenon with RL training, and propose a data filtering technique to mitigate it.

**Reviewer Concerns:**

Since the pass@k degradation has been observed before, the novelty about the reported discovery and analysis appears limited. The proposed mitigation techniques appears not fully justified, with unclear generalization between coding and math.

**Reviewer Scores:**

It is unlikely that the reviewers would have significantly upgraded their scores based on the current reviews,, paper limitations and rebuttal

---

### Decision · Program_Chairs · 2026-01-26

Reject